# SIRT7 as a context-dependent biomarker and therapeutic target: Insights from a pan-cancer study

K.M. Tanjida Islam‡, Shahin Mahmud ‡*

Department of Biotechnology and Genetic Engineering, Mawlana Bhashani Science and Technology University, Tangail, Bangladesh

‡ KMTI and SM equally contributed as first author on this work.
* shahin018mbstu@gmail.com, shahinbge@mbstu.ac.bd

## Abstract

SIRT7 is a member of the sirtuin family and has emerged as a crucial player in cancer biology, with a multifaceted role in both tumor-promoting and tumor-suppressing activities. Despite its importance in connecting NAD+ metabolism with transcriptional regulation, a systematic analysis across multiple cancer types remains underexplored, thereby limiting our understanding of its prognostic value, mutational impact, immune associations, and therapeutic potential. Therefore, this study aims to evaluate the pan-cancer significance of SIRT7 through integrated computational approaches. We employed protein structure modeling, deep neural network-guided protein interaction analysis, cancer hallmark association, gene expression profiling, survival analysis, mutational landscape, immune infiltration assessment, and structure-based drug discovery, combining molecular docking and dynamics simulations. Our deep neural network analyses revealed SIRT7 as a central hub connecting NAD+ metabolism with transcriptional regulation in its sub-network ($R^2$: 0.9839). SIRT7 exhibited differential expression across 17 cancer types, with high expression associated with poor survival in six cancer types; however, it surprisingly correlated with better outcomes in sarcoma. Cancer-specific mutations significantly reduced patient survival and altered the expression of network components. We identified regulatory mechanisms involving five miRNAs and three transcription factors. Therapeutic intervention identified two promising SIRT7 inhibitors (ZINC000150487575 and ZINC000150641215) with superior binding properties compared to the reference inhibitor. This comprehensive pan-cancer analysis of SIRT7 provides a framework for understanding its role in cancer biology and identifies potential therapeutic opportunities for personalized interventions. Our findings have immediate implications for clinical oncology, enabling SIRT7 as a biomarker for patient stratification and a therapeutic target for novel inhibitor development. Targeting SIRT7 may offer new therapeutic strategies for various cancers, particularly those with high SIRT7

**Data availability statement:** All relevant data are within the manuscript and its Supporting Information files.

**Funding:** The author(s) received no specific funding for this work.

**Competing interests:** The authors have declared that no competing interests exist.

expression, as SIRT7 functions in a context-dependent manner in cancer regulation. Further studies are necessary to validate the efficacy of SIRT7 inhibitors and explore their role in therapeutic resistance and disease recurrence.

## 1. Introduction

Despite decades of research and trials of potential new medicines, cancer is still a significant global burden and remains a primary source of death and disability worldwide. [1,2]. Cancer cells undergo genetic and epigenetic alterations that result in uncontrolled growth and the ability to spread to other areas of the body [3]. A recent study estimates that 23.6 million new cases of cancer were diagnosed globally in 2019, leading to 10 million deaths and an estimated 250 million remaining alive with a disability. In comparison to 2010, there has been a significant rise in the incidence, mortality, and disease burden of cancer during the past ten years, with increments of 26.3%, 20.9%, and 16%, respectively. [4].

It has been discovered that SIRT7 has multifaceted functions in cancer, functioning as a tumor suppressor or promoter based on the kind of cancer and the context of the cell [5]. SIRT7 is a member of the sirtuin family that consists of seven members (SIRT1 to SIRT7) [6]. Emerging evidence underscores the critical roles of the seven members of the SIRT family in both health and disease, including regulation of inflammation, metabolism, oxidative stress, and apoptosis [7]. As a result, SIRTs have emerged as promising therapeutic targets across a spectrum of pathological conditions, including cancer, cardiovascular diseases, respiratory disorders, and numerous other diseases [7,8].

In general, SIRT7 plays a pivotal role in both cellular metabolism regulation and genomic stability maintenance in human cell lines. In metabolism, SIRT7 critically regulates glucose and lipid metabolism in white and brown adipose tissues and the liver [9]. For genomic stability, SIRT7 is recruited to DNA damage sites, where it helps recruit repair factors and direct chromatin regulation [10]. SIRT7's absence induces global genomic instability, premature ageing, metabolic dysfunctions, and reduced stress tolerance [11]. Therefore, the normal function of SIRT7 is cellular protection.

In the case of cancer biology, SIRT7 is considered an oncogene, as evidenced by Hepatocellular carcinoma (HCC), where the overexpression of SIRT7 in human HCC samples increased with tumor grade [12]. However, overexpression of SIRT7 reduced lung metastasis in the mouse xenograft model. [8]. Therefore, the role of SIRT7 is associated with both tumor suppression and tumor promotion.

Moreover, the mammalian sirtuin family member SIRT7 is involved in cancer biology through the regulation of DNA damage repair, modulation of oncogenic gene expression, and promotion of chemoresistance in various types of cancer. [13–15]. In addition, SIRT7 regulates numerous biochemical pathways, including both tRNA and rRNA synthesis, which in turn promote enhanced ribosome biogenesis, a process essential for tumor cell growth and proliferation [16].

Remarkably, SIRT7 plays a role in the interaction between tumor-regulatory, chromatin signaling, and metabolic pathways; therefore, SIRT7 may be a possible

therapeutic target for treating epigenetic cancers. [16]. Several studies identified that SIRT7 promotes Melanoma, Lung Cancer, Hepatocellular carcinoma, Thyroid cancer, Gastric Cancer, Colorectal cancer, etc. [12,17,18]. Therefore, SIRT7 can be a key and emerging therapeutic target for treating various types of cancer. However, among sirtuin family members (SIRT1-SIRT7), SIRT7 is still the least studied member till now.

Although SIRT7 has been studied for its role in cancer, a pan-cancer study for SIRT7 is still missing. The multifaceted role of SIRT7, encompassing both tumor-promoting and tumor-suppressing functions, has emerged as a crucial candidate for pan-cancer studies. Besides being a less studied member of the sirtuin family, a pan-cancer study could provide a more comprehensive overview of the cancer associations, mechanisms, and therapeutic potential of SIRT7, which could also aid future research opportunities.

Moreover, over the previous decades, the overall survival rate of cancer patients has not considerably increased because of the implication of chemically modified medications. [19]. As a result, to strengthen the potency of current cancer treatments, new strategies and novel chemoprevention medications are indispensable. Essential sources of natural and synthetic compounds for both revolutionary drugs and cancer treatments are phenolic compounds, nitrogen-containing compounds, organosulfur compounds, carotenoids, and alkaloids found in plants. [20–22]. However, conventional drug discovery techniques typically take more than 12–15 years and cost an average of $1 to $2 billion, spanning from lead identification to clinical trials. [23–25]. In the meantime, *in silico* methods have been gaining a lot of attention recently due to their capability to minimize the time, expense, and labor throughout the drug development process [26].

Therefore, the current study aims to evaluate the pan-cancer potential of SIRT7 and identify potential therapeutic opportunities. The flow diagram of the overall study is shown in Fig 1.

## 2. Materials and methods

### 2.1. Cancer relevance analysis of SIRT7

**2.1.1. 3D model generation and quality assessment of SIRT7.** The 3D model is necessary to generate through computational modeling when an experimentally validated, high-quality X-ray crystallography structure is unavailable [27]. SIRT7 3D structure (UniProt ID: Q9NRC8) was modeled using AlphaFold3 to gain structural insights essential for understanding protein function and potential applications in drug discovery [28]. AlphaFold3 from the AlphaFold server (https://alphafoldserver.com/) was employed for model generation due to its advanced deep learning architecture that accurately predicts protein structure from primary sequence data [29]. Following generation, a comprehensive quality assessment was conducted using multiple validation tools to ensure structural reliability. The SAVES 6.1 suite (https://saves.mbi.ucla.edu/), which includes ERRAT, was utilized to evaluate non-bonded atomic interactions by comparing statistics of atom-atom interactions against reliable high-resolution structures [30]. Additionally, Verify3D was applied to analyze the compatibility of the 3D structure with its amino acid sequence by assigning a structural class based on its location and environment [31]. Furthermore, PROCHECK was employed to validate the stereochemical quality by examining the geometry of residues in comparison to stereochemical parameters from high-resolution structures [32]. In addition, ProSA (Protein Structure Analysis) (https://prosa.services.came.sbg.ac.at/prosa.php) was implemented to evaluate overall model quality through Z-scores that indicate deviation from random conformations [33], while QMEAN (Qualitative Model Energy ANalysis) scoring from the Swiss Model server (https://swissmodel.expasy.org/qmean/) was employed to estimate the global and local quality of the model based on statistical potentials of mean force [34].

**2.1.2. Subcellular localization, immunohistochemistry, expression analysis.** To determine the protein's cellular distribution and tissue-specific expression patterns, subcellular localization and immunohistochemistry data were analyzed using the Human Protein Atlas database (https://www.proteinatlas.org/) [35]. The Human Protein Atlas was employed as it provides comprehensive, publicly available information on protein expression across various human tissues and cell types, generated through systematic immunohistochemistry and microscopy [36].

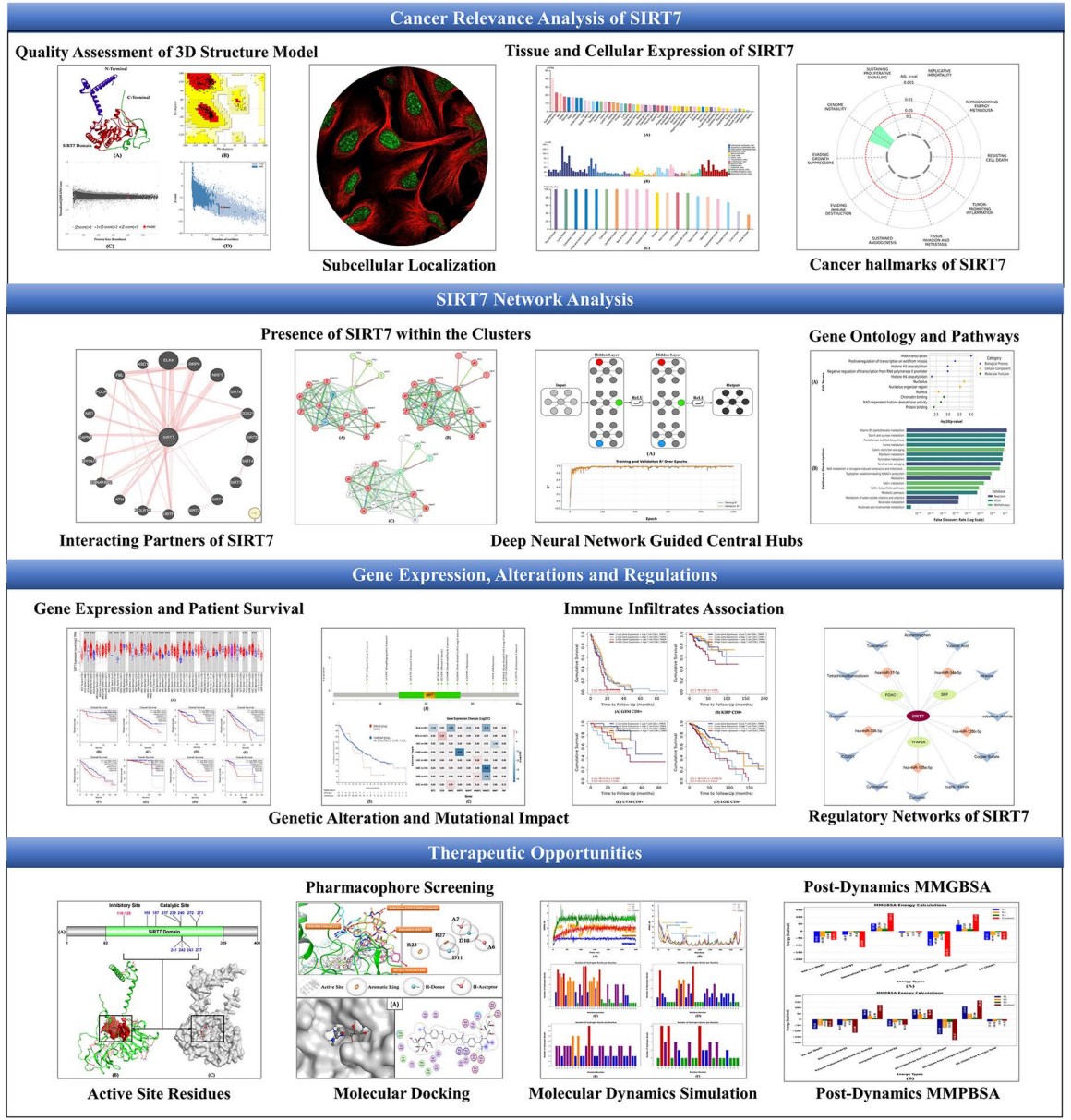

**Fig 1. Flow diagram of the pan-cancer study of SIRT7 comprising the relevance of SIRT7 in cancers, analysis of networks, gene expression, alterations, and regulations of SIRT7, along with identification of therapeutic opportunities.**

Furthermore, to comprehensively characterize the tissue-specific and cellular expression patterns of SIRT7 across human organs and cell types, SIRT7 expression analysis was again conducted using the Human Protein Atlas database.

**2.1.3. Cancer hallmarks analysis of SIRT7.** To investigate the molecular mechanisms and functional implications of SIRT7 in cancer progression, cancer hallmark enrichment analysis was performed using the cancer hallmark database (https://cancerhallmarks.com/) [37]. The analysis was conducted using the integrated cancer hallmark gene set, which comprises 6,763 genes curated from multiple mapping resources, providing a robust framework for understanding the fundamental organizing principles common to various cancers [38]. This methodological approach was specifically

designed to elucidate the association between SIRT7 and specific cancer hallmarks, potentially revealing its role in malignant transformation and progression.

## 2.2. SIRT7 network analysis

### 2.2.1. Identification of interacting partners.
To identify and characterize potential functional interactions between genes and proteins of interest, both gene-gene and protein-protein interaction analyses were performed using well-established bioinformatics platforms. Gene-gene interactions were analyzed using GeneMANIA (https://genemania.org/), a web-based tool that integrates multiple genomic and proteomic datasets to identify networks of functionally associated genes based on co-expression, physical interactions, genetic interactions, shared protein domains, and pathway associations [39]. On the other hand, protein-protein interactions were subsequently investigated using STRING 12.0 (https://string-db.org/), an established database of known and predicted protein-protein interactions that integrates information from experimental repositories, computational prediction methods, and public text collections [40].

GeneMANIA employs a heuristic algorithm to integrate multiple functional association networks and predict gene function using label propagation from a process-specific network [41]. Hence, we have specifically used it to identify physical interactions and co-expression. On the other hand, STRING utilizes an average linkage algorithm for clustering and computes a combined score by integrating probabilities from diverse evidence channels, corrected for random interactions [42]. We have specifically focused on protein-protein interactions, as STRING primarily evaluates experimentally validated and predicted interactions.

Furthermore, the resulting protein-protein interaction network was further analyzed through multiple clustering approaches to identify functional modules within the network. K-means clustering [43], Markov Cluster Algorithm (MCL) clustering [44], and Density-Based Spatial Clustering of Applications with Noise (DBSCAN) clustering [45] were applied to the STRING interaction data to detect densely connected protein communities that potentially represent functional units or biological processes.

### 2.2.2. Automated central hub identification.
Central hub identification is necessary because different measures capture distinct aspects of node importance, providing a comprehensive and robust identification of central hubs in protein-protein interaction networks. Central hubs in the protein-protein interaction network were identified using a NetworkX integrated GCNConv neural network model powered by 12 centrality metrics: Degree, Betweenness, Closeness, Clustering Coefficient, Maximal Clique Centrality (MCC), Maximum Neighborhood Component (MNC), Density of Maximum Neighborhood Component (DMNC), Edge Percolated Component (EPS), Radiality, Stress, BottleNeck, and EcCentricity for robust statistical analysis [46,47]. These hubs are identified through network topology metrics that measure both direct interactions (degree centrality) and indirect influence (betweenness, closeness centralities) within the network. In the context of the SIRT7 Network, a protein is classified as a central hub when it demonstrates significantly high values across multiple centrality metrics (including degree, betweenness, closeness, and clustering coefficients), indicating its importance in maintaining network structure and function.

The GCNConv (graph convolutional neural network) model was developed using the Torch library for tensor operations and neural network architecture. Torch_geometric was employed for implementing the graph neural network to resolve denoising [48]. Additionally, data preprocessing was performed using the StandardScaler from the scikit-learn library [49]. The target and network datasets (generated using STRING 12) utilized to develop the GCNConv model are presented in S1 Table, and the code is accessible at https://github.com/tanjida255/SIRT7Net.git.

Therefore, our deep neural network model, employing GCNConv layers, propagated node features across the 12 network topologies. Node features were normalized using StandardScaler to ensure consistent scaling across diverse molecular data types [49]. The network architecture we developed for our model comprised two GCNConv layers, each followed by a ReLU (Rectified Linear Unit) activation function, which introduces non-linearity to the input data. This non-linearity is essential, as real-world problems are typically non-linear, and such activation functions enable the neural network to

 

learn complex patterns effectively [50]. Furthermore, dropout regularization with a rate of 0.3 was applied between layers to mitigate overfitting. The model's performance was rigorously evaluated using a 60:20:20 train-validation-test split implemented via scikit-learn's train_test_split module. Training and validation losses were monitored over 1,000 epochs to assess the model's effectiveness.

The model's validation methodology incorporates both statistical metrics (MSE, RMSE, $R^2$) and learning curve analysis over 1000 epochs. Internal validation is performed by monitoring training and validation $R^2$ curves for convergence, alongside parallel tracking of loss functions for both datasets. This dual validation approach enables assessment of model stability and detection of potential overfitting during the training process. Finally, the central hub thresholds were determined by ranking the node centrality scores from both the average combined score and the predicted score.

**2.2.3. Gene Ontology and Pathway Analysis.** To elucidate the biological functions and molecular pathways associated with SIRT7, we performed gene ontology and pathway analysis. The gene ontology analysis was conducted using the BioGRID database (https://thebiogrid.org/), a comprehensive repository of protein and genetic interactions that provides curated information on protein function and cellular localization [51]. For pathway analysis, three complementary databases were employed: the Kyoto Encyclopedia of Genes and Genomes (KEGG) database (https://www.genome.jp/kegg/) [52], which offers a systematic analysis of gene functions linking genomic information with higher-order functional information; the Reactome database (https://reactome.org/) [53], a peer-reviewed pathway database of human biological processes that provides detailed molecular events with evidence-based information, and the WikiPathways database (https://www.wikipathways.org/) [54], an open, collaborative platform for capturing and disseminating biological pathway knowledge. Gene sets were analyzed against these databases using standard enrichment analysis methods with a significance threshold of $p < 0.05$.

### 2.3. Analysis of context-dependent role of SIRT7

**2.3.1. Gene expression and patient survival analysis.** To investigate the relationship between gene expression patterns and patient survival outcomes, at first, gene expression analysis was conducted using the Tumor Immune Estimation Resource 2 (TIMER2) (http://timer.cistrome.org/) [55] followed by patient survival analysis through the GEPIA2 platform (http://gepia2.cancer-pku.cn/) [56]. TIMER2 is a comprehensive resource for systematic analysis of gene expression across diverse cancer types and corresponding normal tissues, providing robust statistical comparisons with adjustments for potential confounding factors. On the other hand, GEPIA2, an enhanced web server for analyzing the RNA sequencing expression data from The Cancer Genome Atlas (TCGA) and Genotype-Tissue Expression (GTEx) projects, was employed for survival correlation studies due to its capability to generate customized functional analyses, including correlation assessment, patient survival visualization, and molecular signature identification. The expression data from both databases were normalized using standard algorithms to ensure comparability across different tissue samples and datasets. For survival analysis, Kaplan-Meier curves were generated, where log-rank tests were applied to determine statistical significance [57,58].

**2.3.2. Genetic alteration and mutational analysis.** To comprehensively characterize the genetic landscape and mutational profile of SIRT7 and assess its potential clinical implications, genetic alterations and mutational analyses were performed using established cancer genomics platforms. Genetic alteration analysis was conducted using the cBioPortal database (https://www.cbioportal.org/) [59,60], while mutational analysis was executed through the TIMER2.0 database (http://timer.cistrome.org/) [55]. The cBioPortal platform was selected for genetic alteration analysis due to its comprehensive integration of multidimensional cancer genomics datasets from The Cancer Genome Atlas (TCGA) and other large-scale genomic studies, providing detailed visualization and analysis of complex cancer genomic profiles, including mutations, copy number alterations, and structural variants across multiple cancer types. Additionally, TIMER2.0 was employed for mutational analysis because of its specialized capability to correlate somatic mutations with gene expression.

**2.3.3. Immune infiltrate analysis.** To elucidate the complex interplay between tumor-infiltrating immune cells and cancer progression, immune infiltrate analysis was performed again utilizing the TIMER2.0 (http://timer.cistrome.org/) database [55]. TIMER2.0 was selected for this analysis due to its sophisticated statistical framework that employs multiple immune deconvolution methods to estimate immune cell abundances from gene expression profiles while statistically adjusting for tumor purity, which can significantly confound the interpretation of genomic analyses in cancer tissues. This resource integrates data from The Cancer Genome Atlas (TCGA), spanning over 10,000 samples across 32 cancer types, providing a robust platform for analyzing immune-cancer interactions. The immune infiltrate analysis was executed by querying SIRT7 against the database to assess correlations between gene expression and immune cell infiltration levels, including B cells, CD4＋T cells, and CD8＋T cells. Logrank test p-value was utilized to determine the statistical significance of observed associations.

**2.3.4. Identification of gene regulatory networks.** To decipher the complex regulatory mechanisms governing gene expression patterns in our dataset, we constructed a gene regulatory network focusing on SIRT7-miRNA, SIRT7-Transcription Factors (TFs), and SIRT7-miRNAs interactions. The SIRT7-miRNA-TFs and SIRT7-compounds network analysis were performed using NetworkAnalyst 3.0 (https://www.networkanalyst.ca/), a web-based platform that integrates high-quality data sources and provides visualization tools for biological network analysis [61]. The platform utilizes miRNA interaction data from miRTarBase [62], a curated database of experimentally validated microRNA-target interactions with over 422,000 interactions, and transcription factor (TF) binding information from the ENCODE database [63], which provides genome-wide regulatory element annotations across diverse cell types. NetworkAnalyst 3.0 was implemented with default parameters for network construction, applying a significance threshold of $p < 0.01$ for all predicted interactions. Compound interaction data were obtained from the Comparative Toxicogenomics Database (CTD), which provides curated information about chemical-gene/protein interactions and chemical-disease relationships from the published literature [64].

## 2.4. Therapeutic potential analysis

The therapeutic potential of SIRT7 was investigated to elucidate its relevance in cancer modulation. This analysis was conducted through a sophisticated literature review utilizing the PubMed database [65] and the Google Scholar search engine [66]. Both are well-established platforms for accessing peer-reviewed scientific publications and scholarly articles. The search strategy in PubMed utilized the following keywords: "(SIRT7[Text Word]) AND (overexpression/upregulation[Text Word]) OR (underexpression/downregulation[Text Word]) AND (Cancer[Text Word])" to capture relevant publications through systematic Boolean operators and text word searches. In contrast, the Google Scholar search was employed using Google dorks or an advanced search algorithm with the keywords: "SIRT7" AND "overexpression/upregulation" OR "underexpression/downregulation" AND "Cancer" to identify relevant publications that might not be indexed in PubMed, and therefore minimizing potential information gaps.

**2.4.1. Active site residues and active site identification.** The identification and validation of active site residues are crucial for understanding the structural characteristics and functional domains of a protein. Active site residues were determined through sequence analysis using the NCBI Conserved Domain (CD) database (https://www.ncbi.nlm.nih.gov/Structure/cdd/wrpsb.cgi), a comprehensive resource that enables the identification of conserved functional elements within protein sequences through multiple sequence alignments and domain architecture analysis [67]. The active site pocket of our AlphaFold3-generated SIRT7 structure was subsequently computed using Molecular Operating Environment (MOE), a sophisticated molecular modeling and drug discovery software platform that employs advanced algorithms for protein structure analysis and binding site prediction [68]. The computational predictions were validated by cross-referencing the previously identified conserved residues from the CD database with the spatial coordinates of the computed active site, ensuring the predicted binding pocket contained the evolutionarily conserved functional residues.

**2.4.2. Pharmacophore modeling.** Initially, Maestro's Protein Preparation Wizard was used to prepare the protein-ligand complex by adding hydrogen atoms, assigning bond orders, and minimizing the structure using the OPLS4 force field [69]. Afterward, pharmacophore hypothesis generation was performed to identify essential structural features required for potential inhibitory activity and to facilitate the virtual screening of compound databases. A structure-based pharmacophore hypothesis was generated using Schrödinger's Phase application, incorporating spatial and chemical information derived from the active site cavity and specific active site residues [70]. Phase was selected for its sophisticated pharmacophore perception capabilities and its ability to generate complex hypotheses based on multiple chemical features, including hydrogen bond donors, acceptors, aromatic rings, and hydrophobic regions. The application's advanced algorithms enable the creation of comprehensive three-dimensional pharmacophore models that account for both geometric and chemical complementarity between the ligand and receptor. The generated pharmacophore hypothesis was employed to screen compound databases, filtering molecules based on their spatial arrangement of chemical features that match the established pharmacophore model. The validity of the pharmacophore model was assessed through spatial analysis, confirming that the identified pharmacophoric features corresponded to complementary regions within the active site cavity.

**2.4.3. Molecular docking analysis.** Molecular docking analysis was conducted to evaluate and predict the binding modes and interactions of selected compounds with the target protein. The compounds identified using the pharmacophore features were initially optimized using MOE's QuickPrep tool, which performs essential preprocessing steps, including the addition of hydrogen atoms, generation of appropriate protonation states, calculation of partial charges, and dedicated energy minimization to ensure realistic conformational states of the compounds [68,71]. Subsequently, molecular docking simulations were performed using MOE's induced-fit docking protocol, which accounts for both ligand and receptor flexibility during the binding process. The induced-fit model was selected due to its ability to simulate natural protein-ligand binding events by allowing conformational adjustments in both the ligand and the receptor's binding site residues during the docking process. This sophisticated docking approach incorporates multiple scoring functions to evaluate various aspects of protein-ligand interactions, including hydrogen bonding, van der Waals forces, and desolvation effects.

**2.4.4. ADMET analysis.** The absorption, distribution, metabolism, excretion, and toxicity (ADMET) properties of the candidate compounds were evaluated to assess their drug-likeness and potential pharmacokinetic behavior. The analysis was performed using MOE's ligand properties analysis module, a comprehensive computational tool that employs established algorithms and predictive models to calculate various physicochemical and pharmacokinetic parameters [72]. Additionally, MOE's ADMET prediction system can evaluate multiple drug-relevant properties simultaneously, including molecular weight, logP, hydrogen bond donors and acceptors, polar surface area, and various toxicity indicators based on validated QSAR models. Additionally, hepatotoxicity and carcinogenicity properties of these compounds have been tested using the ProTox 3.0 server, which is an advanced chemical toxicity predictor server, predicts toxicity properties based on molecular similarity and machine learning approaches [73].

**2.4.5. Molecular dynamics simulation.** Molecular dynamics (MD) simulations were performed to evaluate the conformational stability, binding interactions, and dynamic behavior of the protein-ligand complexes under physiologically relevant conditions. The simulations were conducted using AMBER22, a comprehensive suite of biomolecular simulation programs known for its accurate force field parameters and efficient sampling algorithms [74]. The protein system was parameterized using the ff19SB force field [75], selected for its improved backbone and side-chain parameters, while the TIP3P water model [76] was employed for explicit solvation due to its well-validated performance in protein simulations. The system was neutralized and brought to physiological ionic strength (0.15 M) using NaCl ions in an orthorhombic periodic boundary box [77]. Prior to production dynamics, the system underwent a systematic equilibration protocol comprising energy minimization for 20000 steps to relieve unfavorable contacts, followed by NPT equilibration at 300K and NVT equilibration at 1 bar pressure to achieve proper system density and thermal equilibration [78]. Subsequently, production MD simulations were executed for 300 ns to ensure adequate sampling of conformational space. System

stability and binding interactions were analyzed through multiple quantitative metrics: Root Mean Square Deviation (RMSD) to assess overall structural stability, Root Mean Square Fluctuations (RMSF) to identify regions of high mobility, and hydrogen bond occupancy to evaluate the persistence of key binding interactions.

**2.4.6. Post-dynamics MMGBSA and MMPBSA.** The binding free energy calculations were performed using both the Molecular Mechanics Generalized Born Surface Area (MMGBSA) and the Molecular Mechanics Poisson-Boltzmann Surface Area (MMPBSA) approach to quantitatively assess the thermodynamic stability of the protein-ligand complexes and validate the molecular basis of binding interactions. MMGBSA and MMPBSA calculations were executed using AMBER22, which implements robust algorithms for free energy estimation and provides accurate decomposition of energetic contributions [74]. The total binding free energy (ΔGbind) was computed as the difference between the Gibbs free energy of the receptor-ligand complex (Gcomplex) and the sum of the free energies of the unbound receptor (GR) and ligand (GL) components, following the equation:

$$\text{Total binding free energy, } \Delta G_{bind} = G_{complex} - (G_R + G_L) \tag{1}$$

In the equation, $G_{complex}$ = Gibbs free energy of the receptor-ligand complex, $G_R$ = Gibbs free energy of the unbound receptor, and $G_L$ = Gibbs free energy of the ligand.

The calculations were performed on equilibrated trajectories from the molecular dynamics simulations to ensure proper sampling of conformational space and convergence of energetic parameters.

## 3. Results

### 3.1. Cancer relevance of SIRT7

The comprehensive study of cancer relevance of SIRT7 by integrating structural modeling, expression analysis, localization profiling, functional hallmark enrichment, and structural quality assessments provides the multidimensional evidence required to determine whether SIRT7 exhibits cancer-specific activity, druggable structural features, and prognostic or predictive biomarker potential for targeted therapy development.

**3.1.1. Quality assessment of 3D structure model of SIRT7.** The assessment of the overall quality of the 3D structure of the model protein (UniProt ID: Q9NRC8) ensures the reliability and accuracy of the generated protein structure. The ERRAT overall quality factor of the model was 97.24, and the Verify3D score was 86.75%, indicating good compatibility between the 3D model and its amino acid sequence as a good resolution structure generally produces overall quality factor values ≥ 95% and Verify3D score more than 80%. On the other hand, PROCHEK analysis revealed that 90.5% of the amino acid residues are located in the most favored regions, suggesting high-quality stereochemistry as in a high-quality structure, more than 90% of the residues are generally located in the most favored regions. The QMEAN score and Z-score were further validated to assess the quality of the model structure, which were −0.32 (close to 0 is optimum) and −9.06 (model resides in the X-ray crystalographic structural quality region), respectively, providing an estimate of the overall model quality compared to experimentally validated structures of similar size. The **Fig 2**. represents the structural quality assessment of the SIRT7 3D structure that is required for subsequent downstream analysis of binding sites, molecular docking and dynamics simulation.

**3.1.2. Subcellular localization, immunohistochemistry, and expression.** To understand the distribution of SIRT7 within cells, subcellular localization, and immunohistochemistry evidence were evaluated from the Human Protein Atlas database. Subcellular localization studies (**Fig 3A**) revealed that SIRT7 was primarily detected in the nucleoplasm and nuclear speckles, while other cellular compartments showed no detectable presence of the protein [79]. Immunohistochemistry (**Fig 3B**) further confirmed these findings, demonstrating a strong signal for SIRT7 within the nucleus of cells, with a distinct speckled pattern [79]. The red staining highlights the cellular structure, providing context for the observed nuclear localization of SIRT7.

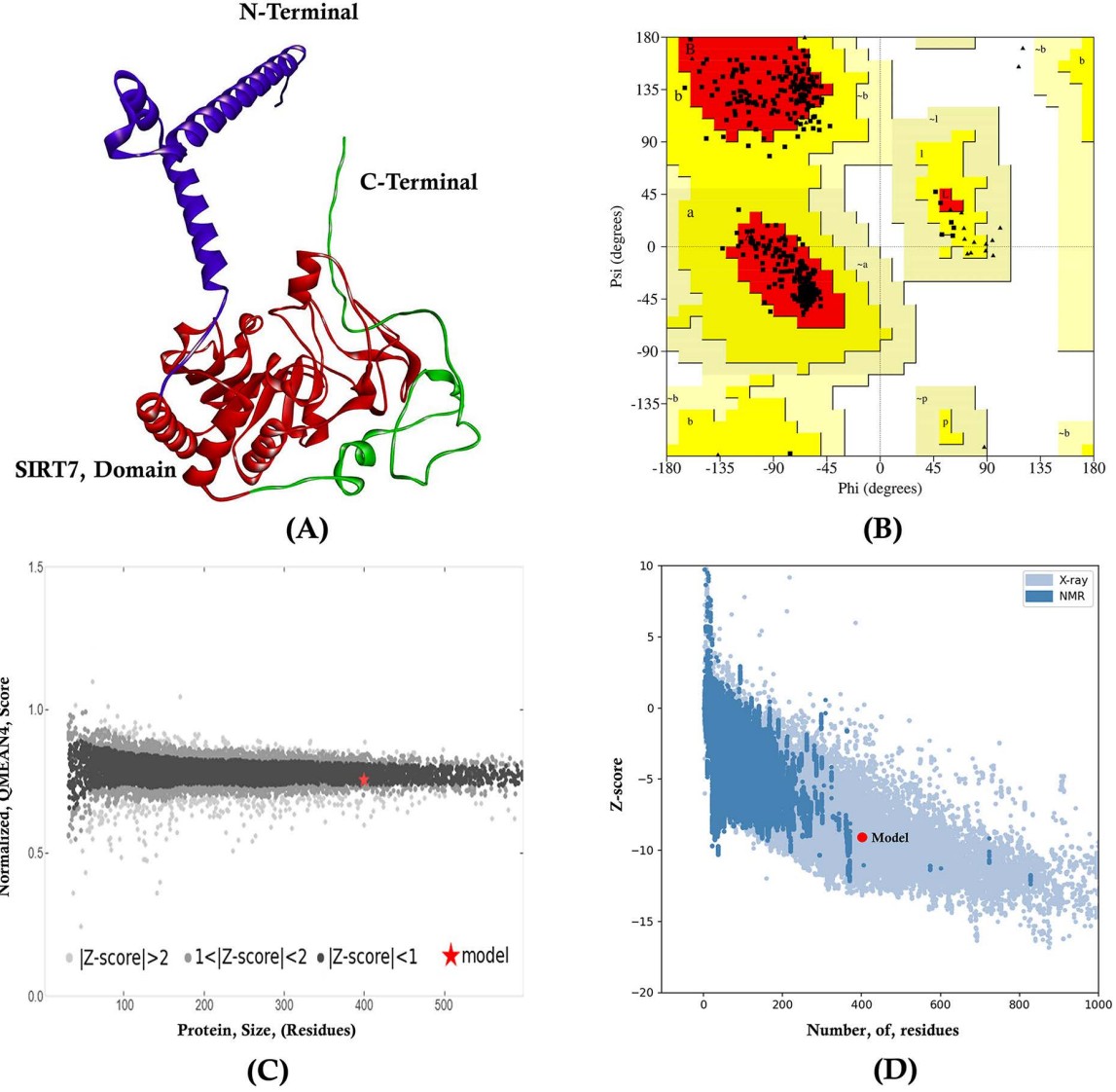

**Fig 2. Quality assessment of the 3D structure model of SIRT7. (A)** Model protein structure (Designed using Discovery Studio 2021), **(B)** Ramachandran Plot (Source: SAVES v6.1), **(C)** Normalized QMEAN plot (Source: Swiss QMEAN), **(D)** Overall model quality plot (Source: ProSA).

Additionally, the analysis of the tissue-specific and cellular patterns of SIRT7 expression is necessary to provide a foundational understanding of its distribution across biological systems. In the tissue expression analysis (**Fig 3C**), the results indicated that *SIRT7* exhibited the highest normalized transcript per million (nTPM) values in the esophagus, while the retina displayed the lowest levels [80]. At the cellular level (**Fig 3D**), glandular epithelial cells showed the highest nTPM values for *SIRT7*, in contrast to adipocytes, which demonstrated the lowest expression [81]. Extending to cancer types (**Fig 3E**), the plot revealed that nearly all thyroid cancer patients had *SIRT7* expression, whereas renal cancer patients exhibited the lowest percentage of expression, highlighting distinct patterns that underscore the variable presence of the protein in health and disease contexts [82].

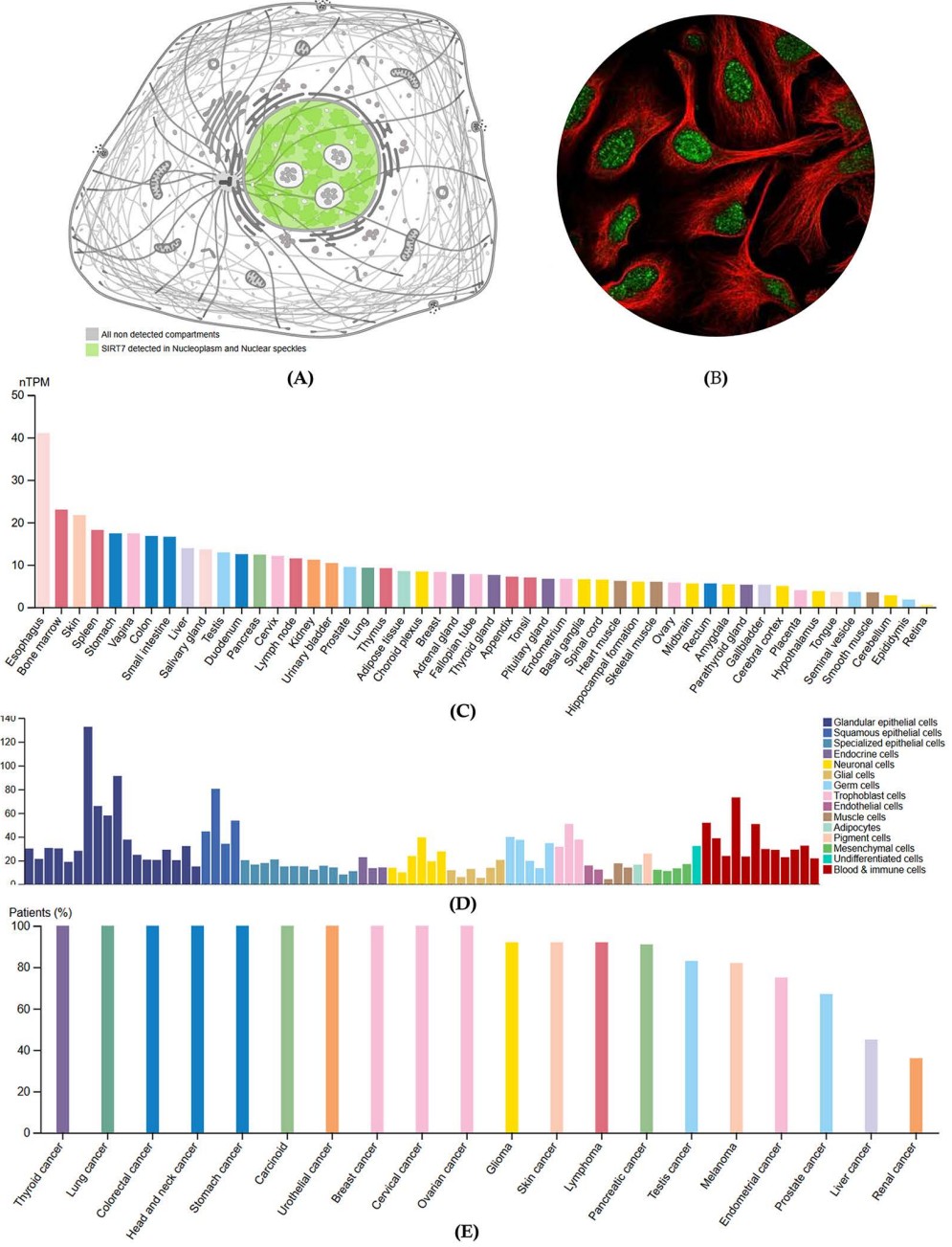

**Fig 3. Cancer relevance analysis of SIRT7. (A)** Subcellular localization of SIRT7. **(B)** immunohistochemistry of SIRT7. Green indicates nuclear regions. **(C)**, **(D)**, **(E)** Expression profile of *SIRT7* across tissues, cell lines, and cancer types, respectively. (Source: Human Protein Atlas).

### 3.1.3. Cancer hallmarks of SIRT7.

To elucidate the role of SIRT7 in cancer development and progression, we evaluated its involvement across multiple cancer hallmarks using adjusted p-value thresholds. The circular plot (Supplementary Fig. S1) revealed that genome instability exhibited a statistically significant association (highlighted in green, adj. p-value < 0.05), suggesting SIRT7's crucial involvement in maintaining genomic stability in cancer cells. In

contrast, several hallmarks, including replicative immortality, reprogramming energy metabolism, resisting cell death, tumor-promoting inflammation, sustaining proliferative signaling, evading immune destruction, sustained angiogenesis, and tissue invasion and metastasis, showed minimal to no significant association (positioned beyond the red dotted significance threshold line, adj. p-value > 0.1).

## 3.2. Networks of SIRT7

Network analysis is essential for identifying central hub proteins and functional modules that could serve as high-confidence biomarker candidates and druggable therapeutic targets. The analysis of topological centrality and pathway connectivity predicts direct clinical relevance and intervention possibilities by focusing on mechanistically significant critical nodes, rather than isolated proteins, thereby guiding the identification of central hubs as biomarkers and therapeutic targets [83,84].

**3.2.1. Interacting partners of SIRT7.** To comprehensively understand the molecular mechanisms and functional networks of SIRT7, we analyzed its interaction patterns through physical interactions, co-expression relationships, and protein-protein interaction networks. The physical interaction analysis (**Fig 4A**) revealed strong direct connections between SIRT7 and multiple partners, with notably intense interactions (indicated by thicker red lines) with ELK4, RRP9, NRF1, and several other nuclear proteins. The co-expression network analysis (**Fig 4B**) demonstrated weaker but consistent correlations among the same set of genes, particularly with SIRT family member (*SIRT6*) and transcriptional regulators (*NSD1*, *CAPN1*). The protein-protein interaction network (**Fig 4C**) uncovered a complex interactome where SIRT7 connects with multiple functional clusters, including NAD+ metabolism-related proteins (NADK2, NAMPT), DNA repair factors (TP53), transcriptional regulators (UBTF, POLI), and other relevant proteins (BST1, CD38, NADSYN1, NMNAT1, NMNAT2, NNMT, NUDT12). The subsequent cluster analysis using three clustering algorithms (K-means, MCL, and DBSCAN) consistently showed SIRT7 clustering with transcriptional regulators (TP53, POLI, UBTF) and NAD+ metabolism enzymes (NADK2, NAMPT, NMNAT1) (S2 Fig).

**3.2.2. Deep neural network guided central hubs.** Performing neural network-guided central hub analysis was necessary to improve the predictive modeling of protein centrality metrics within complex biological networks, utilizing automation, and to identify the most influential hub within a network. The analysis, which integrated various centrality features, revealed that the deep learning model accurately captured the relationships among these features (Table 1, S3A,B,C Fig), achieving training metrics with a Mean Squared Error (MSE) of 0.0001, a Root Mean Squared Error (RMSE) of 0.0073, and an R² value of 0.9990 with minimal loss. These results demonstrated an exceptionally strong fit, indicating that the neural network effectively learned the underlying patterns in the data.

Furthermore, the feature importance analysis (**S4A Fig**), which was utilized to develop our deep neural network model, highlighted that the independent variables: degree centrality (0.146) and betweenness centrality (0.148) were among the most influential predictors, suggesting that these metrics play a crucial role in determining protein centrality. Other features, including closeness centrality (0.132), clustering coefficient (0.125), eccentricity (0.120), average combined score (0.111), and stress (0.111), also contributed significantly to the model's predictive power. The dependent variable is the Predicted Score to rank SIRT7 among its interacting partners. The statistical analysis of 12 centrality metrics is presented in **S4B Fig**.

The proteins analyzed through our deep learning model (**Table 2**), including SIRT7, BST1, CD38, NAMPT, NMNAT1, NMNAT2, NUDT12, NADSYN1, NNMT, UBTF, and POLI, exhibited a range of centrality values. SIRT7, with a high average combined score of 0.9358, had a predicted score of 0.674063. The average combined score encompasses the two most significant interaction patterns: co-expression and experimentally validated interactions. BST1, CD38, NAMPT, NMNAT1, and NMNAT2 showed similar centrality profiles, with predicted scores ranging from 0.666046 to 0.66803. In contrast, proteins like NUDT12 and NADSYN1, with a comparatively lower degree centrality (0.466667) and zero betweenness, had predicted scores of 0.663223 and 0.662102, respectively. NNMT, with a degree of 0.4 and zero

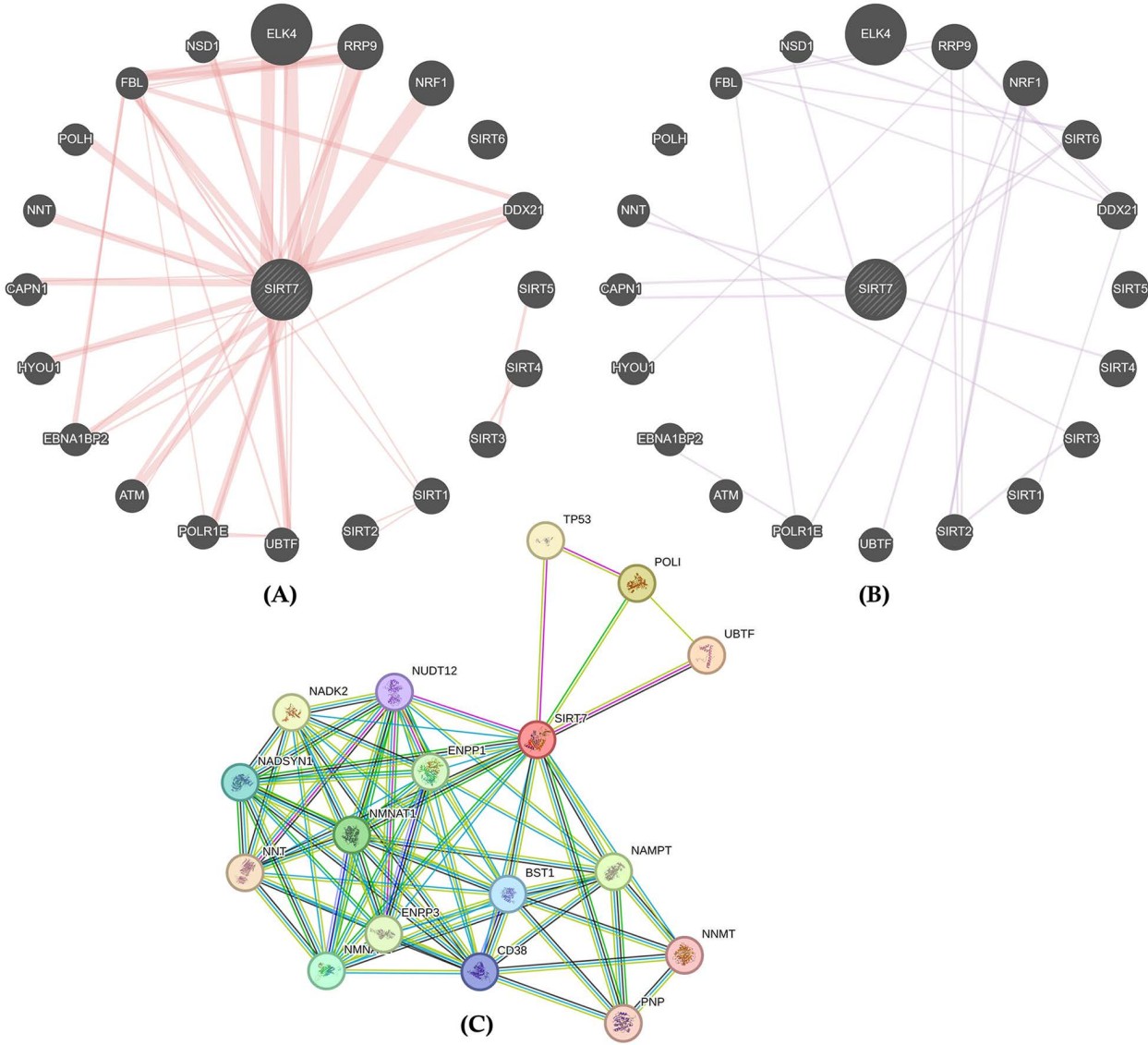

**Fig 4. SIRT7 interacting patterns and partners. (A)** Physical interactions and **(B)** co-expression between *SIRT7* and neighboring genes (Source: GeneMania). **(C)** Protein-protein interaction network of SIRT7 (Source: STRING 12).

**Table 1. Performance of the deep neural network model in calculating hub proteins.**

| Metrics | Training | Validation | Test |
|---|---|---|---|
| R² | 0.9990 | 0.9877 | 0.9839 |
| MSE | 0.0001 | 0.0000 | 0.0008 |
| RMSE | 0.0073 | 0.0061 | 0.0276 |

**Table 2. The most important feature extraction process using the automated deep neural network-based hub proteins prediction model.**

| Proteins | Degree | Betweenness | Closeness | Clustering Coefficient | Average Combined Score | Predicted Score |
|---|---|---|---|---|---|---|
| SIRT7 | 0.666667 | 0.155556 | 0.666667 | 0.6 | 0.9358 | 0.674063 |
| BST1 | 0.533333 | 0.003175 | 0.555556 | 0.928571 | 0.934625 | 0.66803 |
| CD38 | 0.533333 | 0.003175 | 0.555556 | 0.928571 | 0.927875 | 0.667766 |
| NAMPT | 0.533333 | 0.003175 | 0.555556 | 0.928571 | 0.90775 | 0.667001 |
| NMNAT1 | 0.533333 | 0.003175 | 0.555556 | 0.928571 | 0.897 | 0.666656 |
| NMNAT2 | 0.533333 | 0.003175 | 0.555556 | 0.928571 | 0.881375 | 0.666046 |
| NUDT12 | 0.466667 | 0 | 0.512821 | 1 | 0.926 | 0.663223 |
| NADSYN1 | 0.466667 | 0 | 0.512821 | 1 | 0.896143 | 0.662102 |
| NNMT | 0.4 | 0 | 0.47619 | 1 | 0.7725 | 0.65265 |
| UBTF | 0.133333 | 0 | 0.37037 | 1 | 0.967 | 0.577594 |
| POLI | 0.133333 | 0 | 0.37037 | 1 | 0.9585 | 0.577465 |

betweenness, had a predicted score of 0.65265. UBTF and POLI, characterized by lower degree centrality (0.133333) and zero betweenness, but maximal clustering coefficients (1.0), had predicted scores of 0.577594 and 0.577465, respectively.

However, the application of a second layer of shell during deep neural network analysis to the protein network revealed a notable shift in the hub protein composition, with SIRT7 surprisingly absent from the list. This finding suggests that SIRT7's role is specifically tailored to the cluster of proteins comprising SIRT7, BST1, CD38, NAMPT, NMNAT1, NMNAT2, NUDT12, NADSYN1, NNMT, UBTF, and POLI, within which it acts as a central hub. This observation implies that SIRT7's centrality is context-dependent, and its hub status is specific to this subset of proteins. The removal of SIRT7 from the hub list upon the addition of the second shell layer underscores the nuanced and hierarchical nature of protein interactions, highlighting the importance of considering the complex topological structure of protein networks when evaluating the function of hub proteins.

**3.2.3. Gene ontology and pathways.** Functional analysis of gene ontology terms and metabolic pathways was conducted to elucidate the biological context and regulatory mechanisms of SIRT7. Gene ontology analysis (S5A Fig) revealed that SIRT7 functions primarily in epigenetic regulation, specifically in histone deacetylation processes. The protein exhibited NAD-dependent histone deacetylase activity with chromatin binding capability and protein binding properties. Cellular localization was predominantly nuclear, with specific enrichment in the nucleus, nucleolus, and nucleolus organizer region. Regarding biological processes, the protein participated in histone H3 and H4 deacetylation (IDA), negative regulation of transcription from RNA polymerase II promoter, positive regulation of transcription during mitotic exit, and rRNA transcription. Furthermore, Pathway analysis (S5B Fig) identified significant enrichment in nicotinate and nicotinamide metabolism (KEGG: hsa00760, Reactome: HSA-196807) with exceptionally low false discovery rates (5.07e-33 and 9.39e-18, respectively). Additional enrichment was observed in NAD+ biosynthetic and metabolic pathways (WikiPathways: WP3645, WP3644) and tryptophan catabolism leading to NAD+ production (WP4210), consistent with the protein's NAD-dependent enzymatic activity.

## 3.3. Context-dependent Role of SIRT7

The systematic analysis of gene expression patterns, survival outcomes, genetic alterations, immune interactions, and regulatory networks of SIRT7 provides a multidimensional understanding of a gene's biological significance and clinical relevance in cancer progression, which determines its potential as a specific, reliable, and context-dependent biomarker possibility.

### 3.3.1. Gene expression and patient survival.

Performing gene expression and survival analyses is necessary to elucidate the relationship between SIRT7 expression levels and patient outcomes across various types of cancer. Fig. 5A shows the comparison of SIRT7 expression levels between tumor and normal tissues across different cancer types. SIRT7 expression was significantly altered across multiple cancer types, including Bladder Urothelial Carcinoma (BLCA), Breast Invasive Carcinoma (BRCA), Cervical Squamous Cell Carcinoma and Endocervical Adenocarcinoma (CESC), Cholangiocarcinoma (CHOL), Colon Adenocarcinoma (COAD), Esophageal Carcinoma (ESCA), Glioblastoma Multiforme (GBM), Head and Neck Squamous Cell Carcinoma (HNSC), Kidney Chromophobe (KICH), Kidney Renal Clear Cell Carcinoma (KIRC), Kidney Renal Papillary Cell Carcinoma (KIRP), Liver Hepatocellular Carcinoma (LIHC), Lung Adenocarcinoma (LUAD), Lung Squamous Cell Carcinoma (LUSC), Prostate Adenocarcinoma (PRAD), Thyroid Carcinoma (THCA), and Uterine Corpus Endometrial Carcinoma (UCEC), with underexpression observed in Colon Adenocarcinoma (COAD) and Kidney Chromophobe (KICH), and elevated expression detected in the remaining tumor types.

The patient survival results indicated that in BLCA, the survival difference between high and low SIRT7 expression groups was not significant, with a logrank p-value of 0.06 but a high hazard ratio (HR) of 4 was observed (p-value of HR: 0.082) (Fig 5B). In contrast, KIRC showed a survival difference with a logrank p-value of 0.0014 and an HR of 1.6 (p-value of HR: 0.0016) (Fig. 5C), while KIRP exhibited a logrank p-value of 0.034 and an HR of 1.9 (p-value of HR: 0.037) (Fig 5D). LGG demonstrated a survival difference with a logrank p-value of 0.0026 and an HR of 1.7 (p-value of HR: 0.0028) (Fig 5E). LIHC presented with a logrank p-value of 0.0053 and an HR of 1.6 (p-value of HR: 0.0057) (Fig 5F), whereas PRAD had a logrank p-value of 0.014 and an HR of 8.8 (p-value of HR: 0.04) (Fig 5G). SARC showed a logrank p-value of 0.025 and an HR of 0.63 (p-value of HR: 0.026) (Fig 5H), and UCEC had a logrank p-value of 0.019 and an HR of 2.4 (p-value of HR: 0.023) (Fig 5I). These findings collectively indicate variable survival outcomes associated with differing SIRT7 expression levels across cancer types, reflecting the context-dependent, particularly tissue-specific role of SIRT7 in tumor biology.

### 3.3.2. Genetic alteration and mutational impact.

Genetic alteration and mutational impact analysis are essential to comprehend the altered molecular mechanisms underlying SIRT7's involvement across diverse cancer types. Fig 6A illustrates the distribution of specific single nucleotide polymorphisms (SNPs) affecting SIRT7 expression in twelve cancers, including K72N in hepatobiliary cancer, A114V in esophagogastric cancer, Q167E and D234Y in breast cancer, H226Y, R289W, T341I and G375S in melanoma, G246R in head and neck cancer, G268V, and P368A in non-small cell lung cancer, and K395N in colorectal cancer. Notably, unlike single-nucleotide polymorphisms (SNPs), other types of mutations currently lack direct experimental evidence linking them to any specific type of cancer. These mutations are localized predominantly within or near functional domains of the SIRT7 protein, suggesting potential impacts on its activity or regulation. On the other hand, the survival analysis shown in Fig 6B compares patients with altered versus unaltered SIRT7, revealing a trend where the altered group exhibits reduced overall survival probability, with a hazard ratio (HR) of 0.734 (95% CI: 0.376–1.433), implying that mutational status may influence patient prognosis. Fig 6C presents altered SIRT7, which changes gene expression in the protein cluster network of SIRT7, demonstrating variable log2 fold changes (Log2FC) across cancer types for the following genes: BST1, CD38, ENPP1, ENPP3, NAMPT, NMNAT1, NMNAT2, NNMT, and PNP. Notably, substantial downregulation of ENPP3 (Log2FC = −4.23) was observed in COAD, whereas reduced NMNAT2 expression (−4.34) was exhibited in SARC. NMNAT2 (−2.09) also showed decreases in STAD. Other alterations include the downregulation of BST1 in BLCA (−1.10) and the upregulation of CD38 in BRCA (1.20), as well as diminished NNMT in CESC (−1.04). Additionally, in BLCA, ENPP1 (−2.39) and NMNAT2 (−2.53) are also downregulated.

### 3.3.3. Immune infiltrates association.

The relationship between SIRT7 expression, immune cell infiltration, and patient survival across multiple cancer types is often complex. Fig 7A shows that in GBM, high SIRT7 expression combined with high CD8 + T cells was associated with diminished cumulative survival (HR = 1.91, p = 0.0143). Similarly, in KIRP, high SIRT7 expression with elevated CD8 + T cells correlated with reduced survival (HR = 3.36, p = 0.0228)

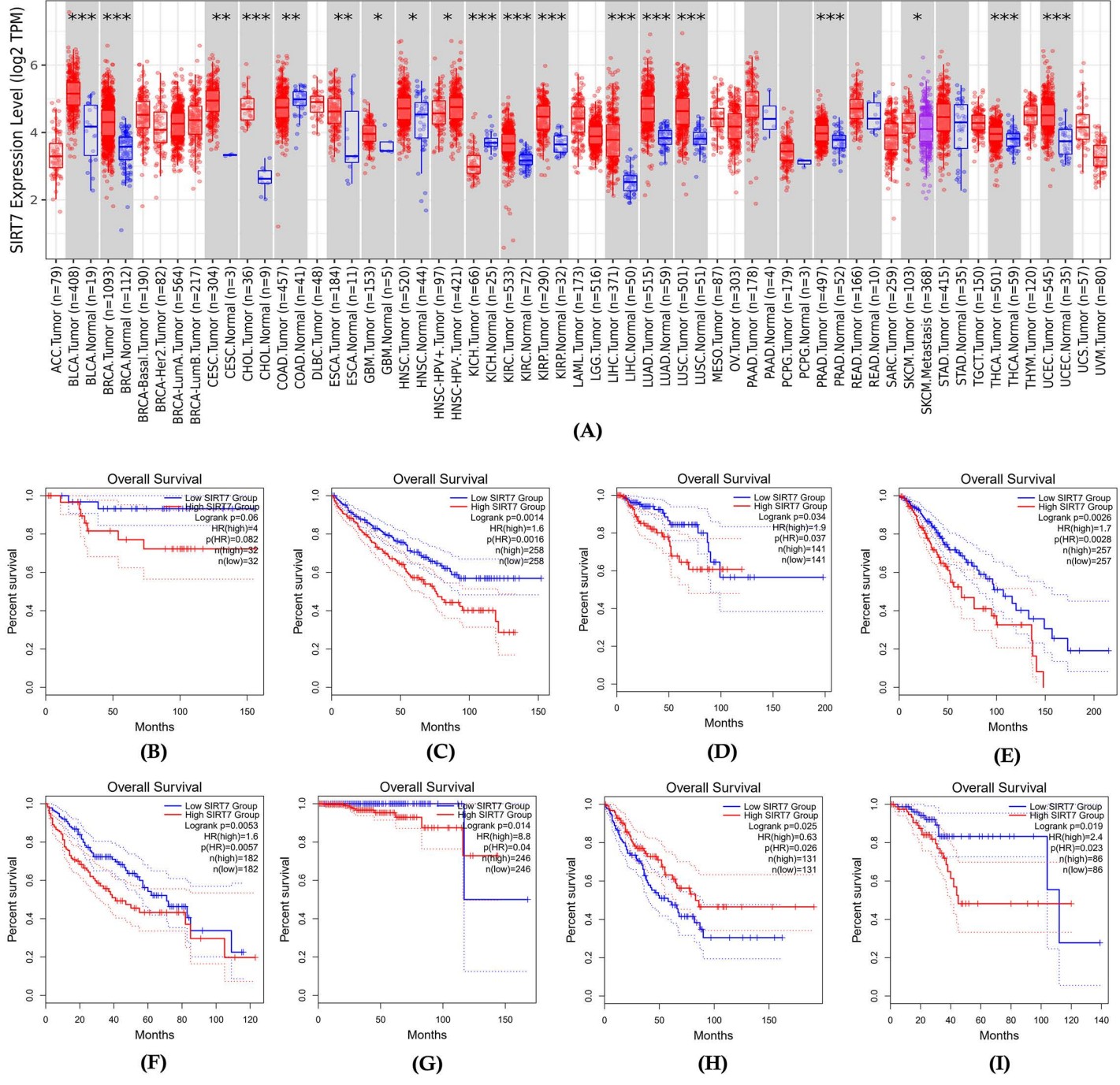

**Fig 5. (A) SIRT7 expression levels (log2 TPM) across multiple cancer types, comparing tumor and normal tissues.** * indicates significant expression, and *** means highly significant expression (Source: TIMER2.0). (B-I) Kaplan-Meier overall survival analyses depicting survival differences between high and low SIRT7 expression groups in (B) BLCA, (C) KIRC, (D) KIRP, (E) LGG, (F) LIHC, (G) PRAD, (H) SARC, and (I) UCEC (Source: GEPIA2).

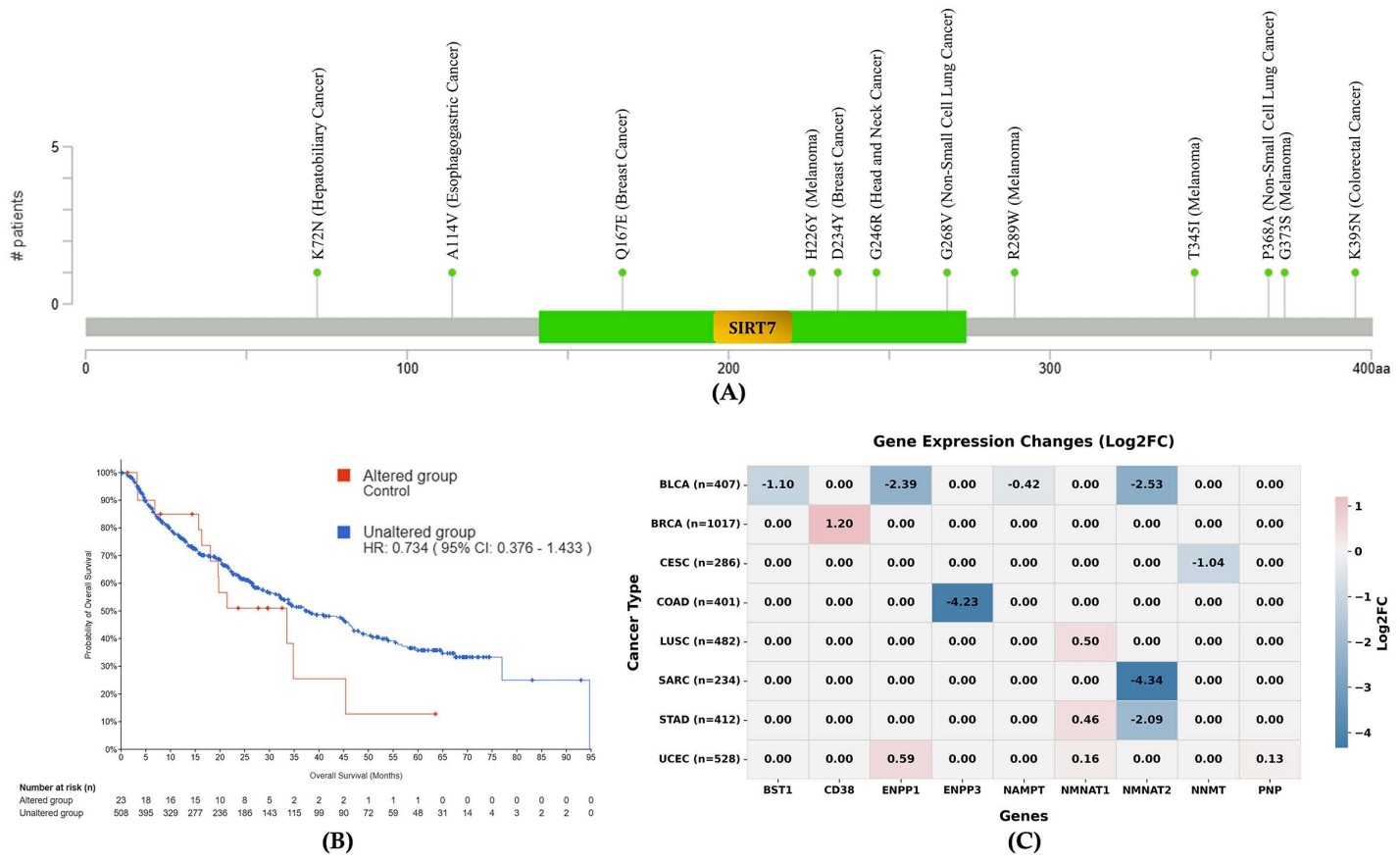

**Fig 6. Genetic alteration analysis. (A)** Mutations that alter SIRT7 expression, and **(B)** SIRT7 survival effects on altered and unaltered groups (Source: cBioPortal). **(C)** Changes in Gene expression of the SIRT7 protein cluster network due to the alteration of SIRT7 (Source: TIMER2).

(**Fig 7B**). In contrast, UVM showed diminished survival with low SIRT7 expression and high CD8+T cells (HR = 4.74, p = 0.0697) (**Fig 7C**). For LGG, high CD4+T cell infiltration was linked to decreased survival for both low SIRT7 expression (HR = 2.87, p = 0.000216) and high SIRT7 expression (HR = 1.9, p = 0.0132) (**Fig 7D**). Finally, in LGG B cells, low SIRT7 expression with high B cell infiltration was associated with lower survival (HR = 1.96, p = 0.0179), while high SIRT7 expression with high B cells also showed diminished survival but to a lesser extent (HR = 1.57, p = 0.059) (**Fig 7E**). These data underscore the nuanced influence of SIRT7 expression in modulating immune infiltrate-related survival outcomes in a tumor-specific manner. These findings suggest a complex and context-dependent relationship between SIRT7 expression, the presence of immune cells, and patient prognosis across various tumor types.

### 3.4. Regulatory networks of SIRT7

Elucidating the regulatory networks of SIRT7 is crucial for understanding its multifaceted control and identifying novel intervention points for therapeutic development. To characterize the regulatory networks influencing SIRT7, we examined its modulation by various drugs and compounds, microRNAs, and transcription factors. The analysis identified multiple drugs and compounds, including Acetaminophen, Valproic Acid, Atrazine, Cobaltous Chloride, Copper Sulfate, Cupric Chloride, Curcumin, Cyclosporine, ICG 001, Quercetin, Tetrachlorodibenzodioxin, Tunicamycin, as potential modulators of SIRT7 activity. Additionally, five microRNAs: hsa-miR-17-5p, hsa-miR-34a-5p, hsa-miR-125a-5p, hsa-miR-125b-5p, and

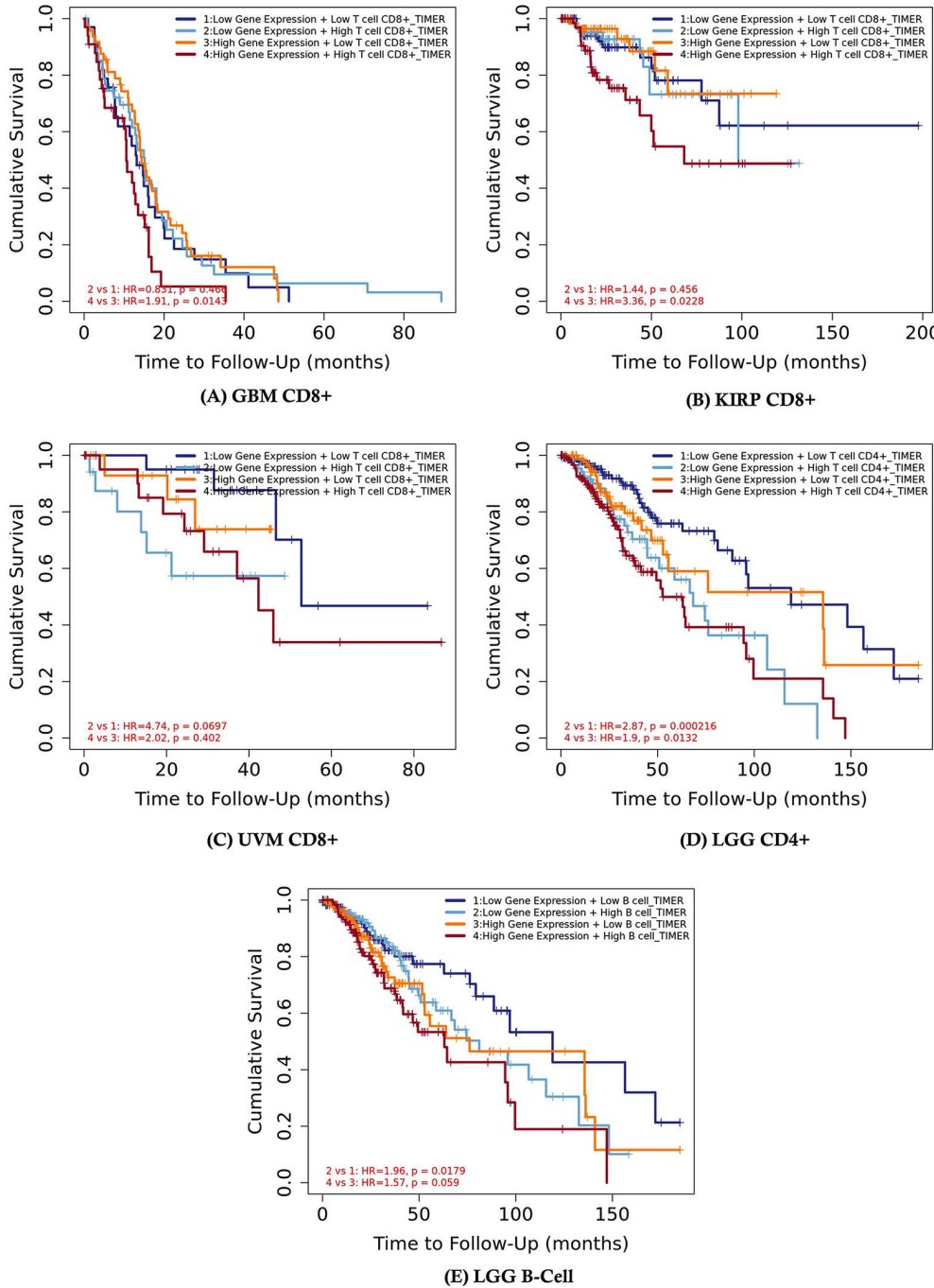

**Fig 7. Survival analysis of SIRT7 expression and immune cell infiltration impact on patient outcomes in (A) Glioblastoma Multiforme (GBM) CD8+, (B) Kidney Renal Papillary Cell Carcinoma (KIRP) CD8+, (C) Uveal Melanoma (UVM) CD8+, (D) Lower Grade Glioma (LGG) CD4+, and (E) Lower Grade Glioma (LGG) B-Cell Populations (Source: TIMER2).**

hsa-miR-335-5p were found to repress or inhibit SIRT7 expression, indicating layers of post-transcriptional regulation. The transcription factors FOXC1, SRF, and TFAP2A were identified as upstream regulators of SIRT7, suggesting precise transcriptional control. The integrated network highlighting the complex and multifaceted regulation of SIRT7 through chemical, transcriptional, and post-transcriptional mechanisms is presented in S6 Fig.

## 3.5. Therapeutic opportunities

The role of SIRT7 in cancer progression has been a subject of investigation, and understanding its mechanisms of action is crucial for identifying potential therapeutic targets. SIRT7 has been found to promote various types of cancer, including melanoma, non-small cell lung cancer, gastric cancer, hepatocellular carcinoma, thyroid cancer, colorectal cancer, epigenetic cancer, anaplastic thyroid cancer, and gallbladder cancer, through distinct pathways (Table 3). In melanoma, SIRT7 promotes tumor cell survival and immune evasion via UPR activation, whereas in non-small cell lung cancer, it facilitates G1/S phase transition, EMT, and activates AKT and ERK1/2 signaling. Similarly, SIRT7 has been shown to promote growth and inhibit apoptosis by epigenetically inhibiting miR-34a in gastric cancer and enhance tumorigenesis in colorectal cancer. These findings indicate that SIRT7 plays a multifaceted role in cancer progression, and its dysregulation contributes to the development and progression of various cancer types.

**3.5.1. Active site residues.** The structural analysis and mapping of SIRT7's active site residues, spanning amino acids 1–400, were crucial for understanding its molecular architecture and functional mechanisms. The SIRT7 domain structure (Fig 8A) revealed two distinct functional regions: an inhibitory site (residues 116–128) [90] and a catalytic site containing key residues (169, 187, 237, 239, 240, 272, 273) [91], along with additional regulatory positions (241, 242, 243, 277), which are also potential inhibitory sites [92,93].

The study identified two cyclic tripeptides as potent inhibitors of human SIRT7 deacetylase activity, exhibiting low µM level inhibitory potency and pan-inhibition against the deacylase activities of five human sirtuins, including SIRT1/2/3/6/7, marking the first reported potent SIRT7 inhibitors to date.

**Table 3. Types of cancer promoted by SIRT7.**

| Cancer Type | Mechanism of Action of SIRT7 in Cancer Progression | References |
|---|---|---|
| Melanoma | Promotes tumor cell survival and immune evasion via UPR activation. | [17] |
| Non-Small Cell Lung Cancer (NSCLC) | Facilitates G1/S phase transition, EMT, and activates AKT and ERK1/2 signaling. | [18] |
| Gastric Cancer | Promotes growth and inhibits apoptosis by epigenetically inhibiting miR-34a. | [12] |
| Hepatocellular Carcinoma | Promotes tumor cell growth. | |
| Thyroid Cancer | Promotes carcinogenesis driven by PTEN deficiency. | |
| Colorectal Cancer | High levels of expression enhance tumorigenesis. | |
| Epigenetic Cancer | Promote the increased ribosome biogenesis necessary for tumor cell growth and proliferation | [16] |
| Breast Cancer | Downregulation promotes breast cancer metastasis via LAP2α-Induced chromosomal instability | [85] |
| Prostate Cancer | Deacetylations of histone H3 at lysine 18 suppress tumor suppressor genes and promote cancer cell proliferation and survival. | [86] |
| Renal Cancer | SIRT7-mediated deacetylation of CHD1L amplifies HIF-2α-dependent signal that drives renal cell carcinoma progression | [87] |
| Anaplastic Thyroid Cancer | Promotes by regulating the desuccinylation of KIF23 gene | [88] |
| Gallbladder Cancer | Reduces activation of the NF-κB signaling pathway | [89] |

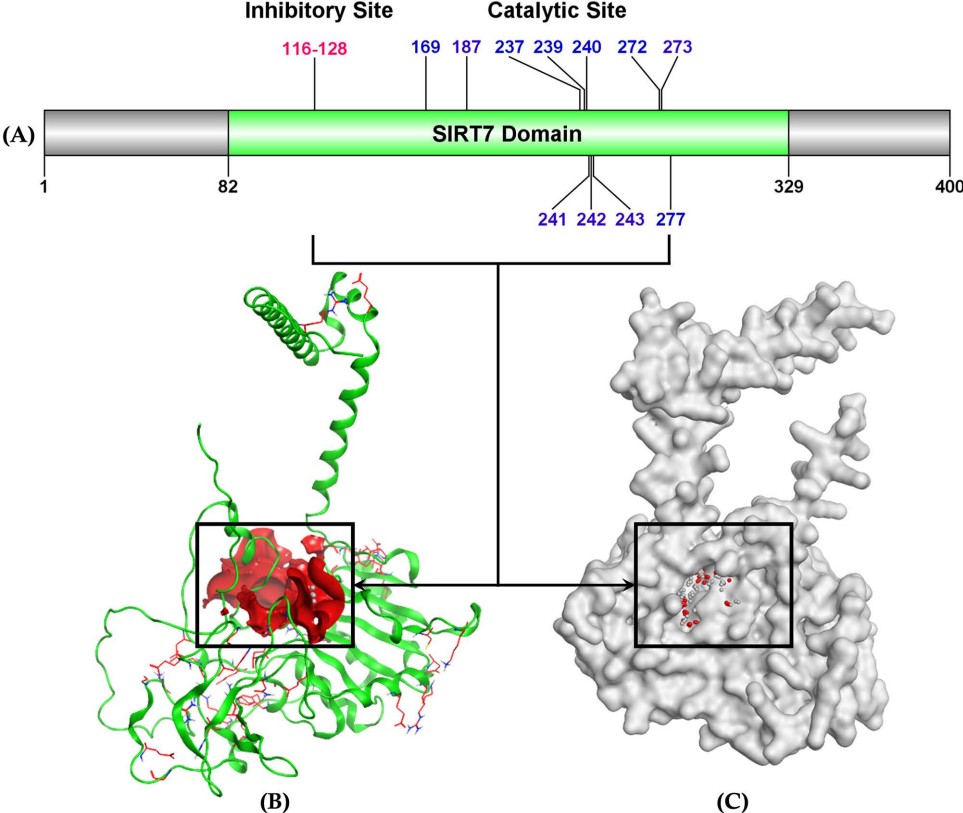

**Fig 8. Localization of active site residues within the SIRT7 domain, highlighting the inhibitory site (residues: 116–128) and the catalytic site (residues: 169, 187, 237, 239–243, 240, 272, 273, 277). (A)** Schematic representation of the SIRT7 domain indicating key residue positions. **(B)** Ribbon diagram of SIRT7 with the inhibitory site surface highlighted in red. **(C)** Surface representation of SIRT7 emphasizing the catalytic site residues in red within the active pocket (Designed using MOE 2019 release).

Analysis of the crystal structure (Fig 8B and C), demonstrated 53 active site residues: GLU77, GLY107, ALA108, GLY109, ILE110, SER111, THR112, ALA113, SER115, ILE116, PRO117, ASP118, TYR119, ARG120, GLY121, PRO122, ASN123, GLY124, THR146, THR148, GLN167, ASN168, CYS169, HIS187, ILE236, VAL237, HIS238, PHE239, GLY240, GLU241, GLY268, SER269, SER270, LEU271, LYS272, VAL273, LEU274, LYS276, TYR277, VAL296, ASN297, LEU298, GLN299, TRP300, GLY313, LYS314, CYS315, ASP316, TRP385, GLY387, ARG388, GLY389, CYS390 created a sophisticated binding site. The catalytic core, highlighted by the black boxes in both representations, showed the region of these critical residues forming the active site pocket. The structural data indicated that the SIRT7 domain possessed a complex arrangement of amino acids that formed distinct microenvironments within both the inhibitory and catalytic sites, with the surface representation particularly emphasizing the spatial organization and accessibility of these functional regions. The catalytic domain exhibited conserved residues, including CYS169, HIS187, and HIS238, which formed the core catalytic triad, while the inhibitory site contained a distinctive sequence from GLU77 to CYS390. The structural arrangement revealed that the domain comprised a mixture of charged, polar, and nonpolar residues, indicating a sophisticated mechanism for substrate recognition and catalysis.

**3.5.2. Pharmacophore Features.** The development of pharmacophore models based on active site information is crucial for rational drug design and virtual screening campaigns aimed at identifying feature-specific lead compounds. The pharmacophore model was constructed utilizing key active site residues including CYS169, HIS187, VAL237, PHE239,

GLY240, GLU241, ARG242, GLY243, LYS272, VAL273, and TYR277, which led to the identification of seven critical features: two aromatic rings (at coordinates 0.4, −2.9, 2.2 and 4.1, 3.7, 0.9), one negative ionizable group (at 1.6, 1.2, 1.4), two hydrogen donors (at 8.5, 6.2, 1.2 and 2.7, 6.3, 1.5), and two hydrogen acceptors (at 10.1, 6.4, 5.1 and 8.4, 4.9, −0.5) (S2 Table). Fig 9 revealed the spatial arrangement of these features, with R23 and R27 representing aromatic rings, D10 and D11 as hydrogen donors, and A6 and A7 as hydrogen acceptors, all positioned strategically to complement the binding pocket topology. The subsequent virtual screening campaign across eight diverse chemical databases (S3 Table), encompassing over 194 million compounds, yielded 52 hits that matched the established pharmacophore features. The most significant number of hits was obtained from ChEMBL (30 compounds), followed by ZINC (8 compounds) and IMPPAT (5 compounds), while specialized databases such as ChemSpace, MCULE Ultimate, and MolePort contributed one hit each.

**3.5.3. Molecular docking.** Molecular docking analysis was essential to evaluate the binding affinity and interaction patterns of the identified compounds with the target protein's active site. Performing molecular docking is necessary to investigate the specific interactions and binding affinities of candidate compounds at the inhibitory site of the SIRT7

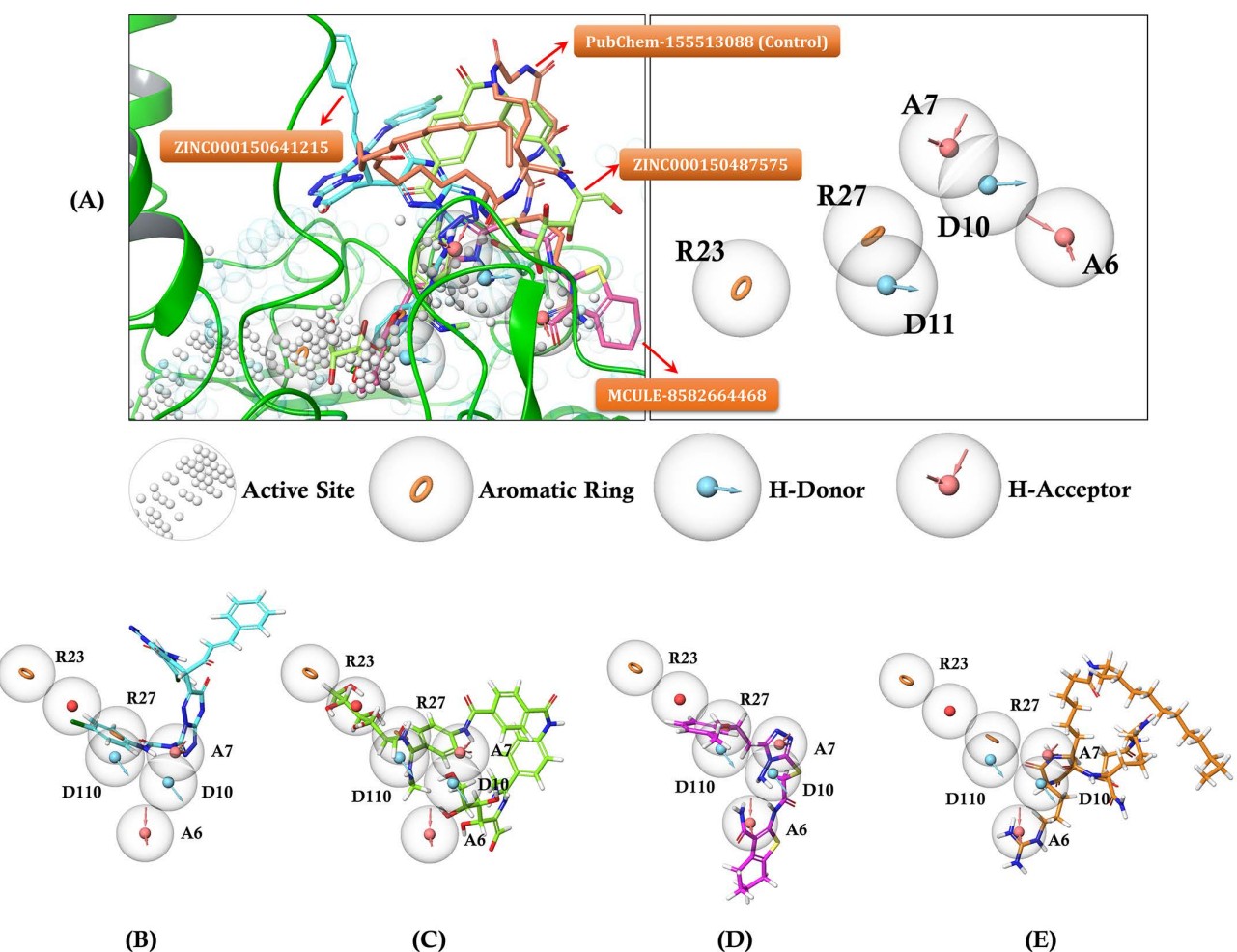

**Fig 9. Mapping pharmacophore features. (A)** Superimposed view of pharmacophore features mapping within the active site of the target protein. **(B)** ZINC000150641215, **(C)** ZINC000150487575, **(D)** MCULE-8582664468, and **(E)** CID: 155513088 (Control). (Designed using Schrodinger 2024.2 release).

domain, thereby elucidating their potential inhibitory mechanisms. The docking analysis revealed that ZINC000150487575 exhibited the most favorable binding affinity with a docking score of −12.5225 kcal/mol. This compound formed a critical hydrogen bond with ARG120, acting as a backbone donor, while GLY121 and ASP118 contributed through hydrophobic interactions, further stabilizing the binding at the inhibitory site. For ZINC000150641215, the docking score was −10.6249 kcal/mol, where ASN123 served as a backbone hydrogen bond donor and PRO122 as a sidechain donor, both involved in hydrogen bonding; ASP118, PRO122, and ASN123 also participated in hydrophobic interactions, indicating a multifaceted binding profile. MCULE-8582664468 demonstrated a docking score of −10.5185 kcal/mol but did not form any hydrogen bonds at the inhibitory site; instead, its binding was dominated by hydrophobic contacts with TYR119, ASP118, and ARG120. The experimentally validated inhibitor, PubChem CID: 155513088 [90], showed a docking score of −9.71424 kcal/mol and formed hydrogen bonds with ASN123 through both backbone and sidechain donors. Additionally, this compound engaged a broader array of residues, including PRO117, ASP118, TYR119, ARG120, GLY121, PRO122, ILE116, and others in hydrophobic interactions, highlighting an extensive interaction network. Across all compounds, the amino acids involved in binding, such as ARG120, ASP118, ASN123, and PRO122, reside consistently within the SIRT7 domain, underscoring the specificity of inhibitor binding at this site. The docking results and their interactions are presented in Table 4 and Fig 10. A complete list of docking complexes, including their binding energies and toxicity profiles, is provided in S4 Table.

**3.5.4. Toxicity analysis.** The assessment of toxicity properties serves as a critical checkpoint for potential pharmacokinetic challenges and safety concerns prior to experimental validation. Table 5 presents the analysis of four compounds, which revealed molecular weights ranging from 500.65 to 836.08 Da, with PubChem-155513088 exhibiting the highest molecular mass and MCULE-8582664468 the lowest. Among the evaluated compounds, MCULE-8582664468 contained an aromatic amine as a potentially toxic group, while the remaining compounds exhibited no concerning structural elements. The toxicity predictions demonstrated favorable outcomes, as all compounds were classified as inactive for both hepatotoxicity and carcinogenicity parameters except ZINC000150641215, which is classified as an active hepatotoxic, but still, the hepatotoxic and carcinogenic probability score is around only 0.57. These computational predictions provided initial insights into the drug-likeness and safety profiles of the compounds, with three out of four compounds showing the absence of toxic structural moieties while maintaining reasonable molecular weights compared to the control experimentally validated inhibitor.

**3.5.5. Molecular dynamics simulation.** Molecular docking provides insights into binding poses in static conditions, but it doesn't capture the dynamic nature of protein and ligand. On the other hand, molecular dynamics simulation provides essential insights into the structural stability, conformational flexibility, and interaction patterns of

**Table 4. Molecular docking analysis to capture interacting amino acids with compounds in the most suitable binding pose.**

| Compound ID | Docking Score (kcal/mol) | Amino Acids |
| --- | --- | --- |
| ZINC000150487575 | −12.5225 | PHE239, HIS238, ILE236, HIS187, SER270, GLN167, **ARG120**, **GLY121**, GLU77, **ASP118**, ASP316, THR112 |
| ZINC000150641215, PubChem CID:136632847 | −10.6249 | THR112, SER115, GLY109, LYS314, GLU77, CYS315, GLY266, ASN268, ASN297, GLN299, LEU298, **ASP118**, **PRO122**, **ASN123** |
| MCULE-8582664468 | −10.5185 | **TYR119**, **ASP118**, **ARG120**, GLN299, ASN297, HIS187, GLN167, ASN168, SER270, GLY268, GLY109, GLU77, ALA113, THR112, LEU298, CYS315 |
| PubChem CID: 155513088 (Control Experimentally Validated Inhibitor of SIRT7) | −9.71424 | THR112, **PRO117**, **TYR119**, SER269, ASN168, SER270, **ARG120**, ALA108, GLY107, GLY109, LYS314, GLU77, **ASP118**, **GLY124**, **GLY121**, **ASN123**, **PRO122**, **ILE116**, CYS315, GLN167, ASN297, GLN299, LEU298 |

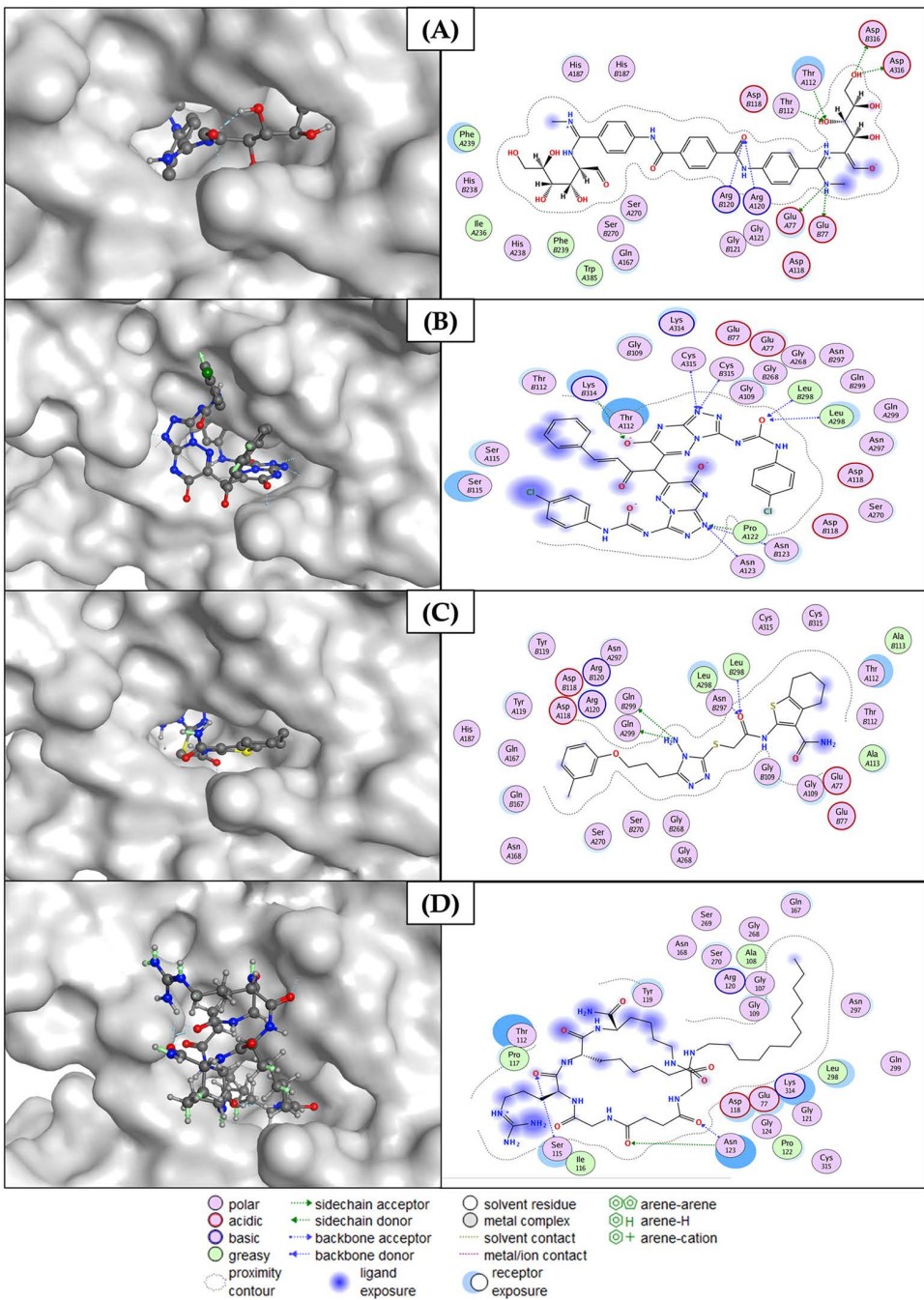

**Fig 10. Molecular docking analysis of candidate inhibitors reveals key hydrogen bonding and hydrophobic interactions within the SIRT7 inhibitory domain. (A)** ZINC000150487575, **(B)** ZINC000150641215, **(C)** MCULE-8582664468, and **(D)** PubChem-155513088 (Control) (Designed using MOE 2019 release).

protein-ligand complexes over time under simulated dynamic conditions similar to those found in the human body environment. This enables a deeper understanding of their dynamic behavior under physiological conditions. **Fig 11A** shows the analysis of RMSD (Root Mean Square Deviation) that revealed distinct stability profiles among the

**Table 5. Toxicity profiling of selected compounds: molecular weight, toxicophores, hepatotoxicity, and carcinogenicity assessment.**

| ID | Molecular Weight | Toxic Group | Hepatotoxicity (Probability) | Carcinogenicity (Probability) |
|---|---|---|---|---|
| ZINC000150487575 | 753.79 | No | Inactive (0.79) | Inactive (0.68) |
| ZINC000150641215 PubChem-136632847 | 749.50 | No | Active (0.57) | Inactive (0.58) |
| MCULE-8582664468 | 500.65 | Aromatic Amine | Inactive (0.57) | Inactive (0.51) |
| PubChem-155513088 | 836.08 | No | Inactive (0.87) | Inactive (0.58) |

complexes, with S1 (ZINC000150487575) exhibiting the lowest average RMSD of 1.5882Å, indicating enhanced structural stability compared to S2 (ZINC000150641215) (2.9337Å), S3 (MCULE-8582664468) (4.1824Å), and the control (PubChem-155513088) (2.8052Å). The RMSD plos indicated that after initial equilibration, all systems reached relative conformational stability after 160 ns, though S3 maintained consistently higher RMSD values throughout the simulation period. Furthermore, the RMSF (Root Mean Square Fluctuation) analysis highlighted regions of notable flexibility, particularly at residue ALA41, which demonstrated peak fluctuations in both S1 (4.5921Å) and S2 (4.4404Å) systems (**Fig 11B**). The residue-wise fluctuation patterns, as depicted in the RMSF plots, revealed several regions of heightened mobility, with notable peaks corresponding to loop regions and termini, while the core structural elements maintained relatively lower fluctuations across all systems, suggesting preserved structural integrity despite dynamic movements in specific regions. The hydrogen bonding analysis unveiled specific interaction patterns unique to each complex: S1 displayed strong hydrogen bonding networks involving residues 182, 185, 187, 188, 190, and 198 (**Fig 11C**), while S2 showed concentrated hydrogen bonding at residues 197 and 199 (**Fig 11D**). S3 exhibited a more focused hydrogen bonding pattern centered at residue 193 (**Fig 11E**), and the control system demonstrated strong hydrogen bonding interactions at residues 171, 178, and 180 (**Fig 11F**).

**3.5.6. Post-dynamics MM-GBSA and MM-PBSA.** Post-dynamics energetic analysis using MM-GBSA and MM-PBSA methods provides crucial insights into the binding free energies and thermodynamic stability of protein-ligand complexes, offering a more refined assessment of their interaction patterns. In the MM-GBSA calculations (**Fig 12A**), the energy components were systematically analyzed. The Van der Waals interactions showed values of −79.52, −41.93, −13.61, and −61.31 kcal/mol for S1, S2, S3, and the control, respectively. The electrostatic energy contributions were computed as −26.47, −5.61, −5.18, and −115.06 kcal/mol, while the generalized Born energy terms were 52.50, 21.54, 17.37, and 125.31 kcal/mol for the respective systems. Surface energy calculations yielded −8.63, −4.47, −1.73, and −7.98 kcal/mol. The gas phase ΔG showed values of −105.99, −47.56, −21.79, and −176.37 kcal/mol, while solution phase ΔG values were 43.87, 17.07, 15.64, and 117.13 kcal/mol, ultimately resulting in total binding free energies (ΔG Total) of −62.22, −30.48, −6.15, and −59.24 kcal/mol for S1, S2, S3, and control systems, respectively. The MM-PBSA analysis (**Fig 12B**) incorporated additional energy components: the Van der Waals and electrostatic energies remained consistent with those of MM-GBSA, while the Poisson-Boltzmann electrostatic energy showed values of 84.41, 30.25, 13.38, and 125.92 kcal/mol. Nonpolar solvation energy contributions were −54.42, −26.77, −9.42, and −45.13 kcal/mol, and dispersion energy terms were 86.48, 44.73, 13.53, and 82.85 kcal/mol. The gas and solution phase ΔG calculations resulted in final total binding free energies of −17.32, −18.95, −5.31, and −13.43 kcal/mol for S1, S2, S3, and control systems, respectively. In the case of MMGBSA, the ΔG Total of S2 was less than that of S1 and the control. On the other hand, in the case of MMPBSA, S2 was higher than both S1 and control, indicating a strong binding free energy of S2. However, both approaches consistently demonstrated more favorable binding energetics for S1 and S2 compared to S3, with energy decomposition revealing that Van der Waals interactions and electrostatic forces make the dominant contributions to the overall binding stability.

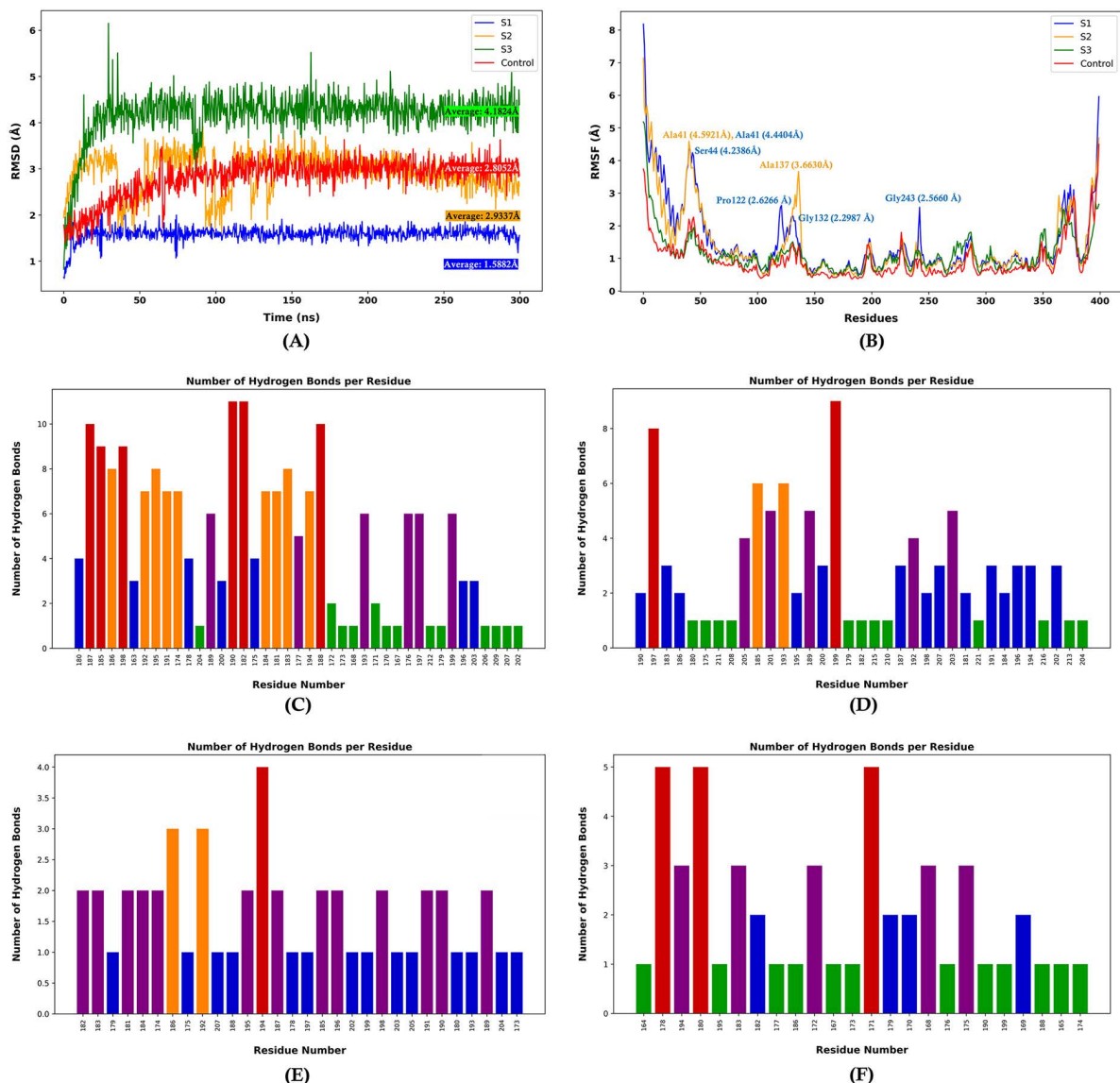

**Fig 11. Molecular dynamics simulations. (A)** RMSD, **(B)** RMSF, **(C)**, **(D)**, **(E)**, and **(F)** Hydrogen bonding occupancy of amino acid residues for compound S1, compound S2, compound S3, and control inhibitor, respectively (Designed using Matplotlib 3.10.0).

## 4. Discussion

SIRT7 is associated with various types of cancer, and the molecular mechanisms of SIRT7 have been evaluated for a wide range of cancers [85,86,94]. However, a comprehensive pan-cancer analysis of SIRT7 remains lacking. Therefore, to evaluate the influence of SIRT7 in regulating cancers, along with its expression patterns in pan-cancer scenarios, the significance of molecular interactions, prognostic value, and therapeutic potential across diverse cancer types is prioritized in our study.

Our comprehensive pan-cancer analysis of SIRT7 reveals several significant findings with implications for cancer biology and therapeutic development. The structural quality assessment confirmed the reliability of our SIRT7 3D model with

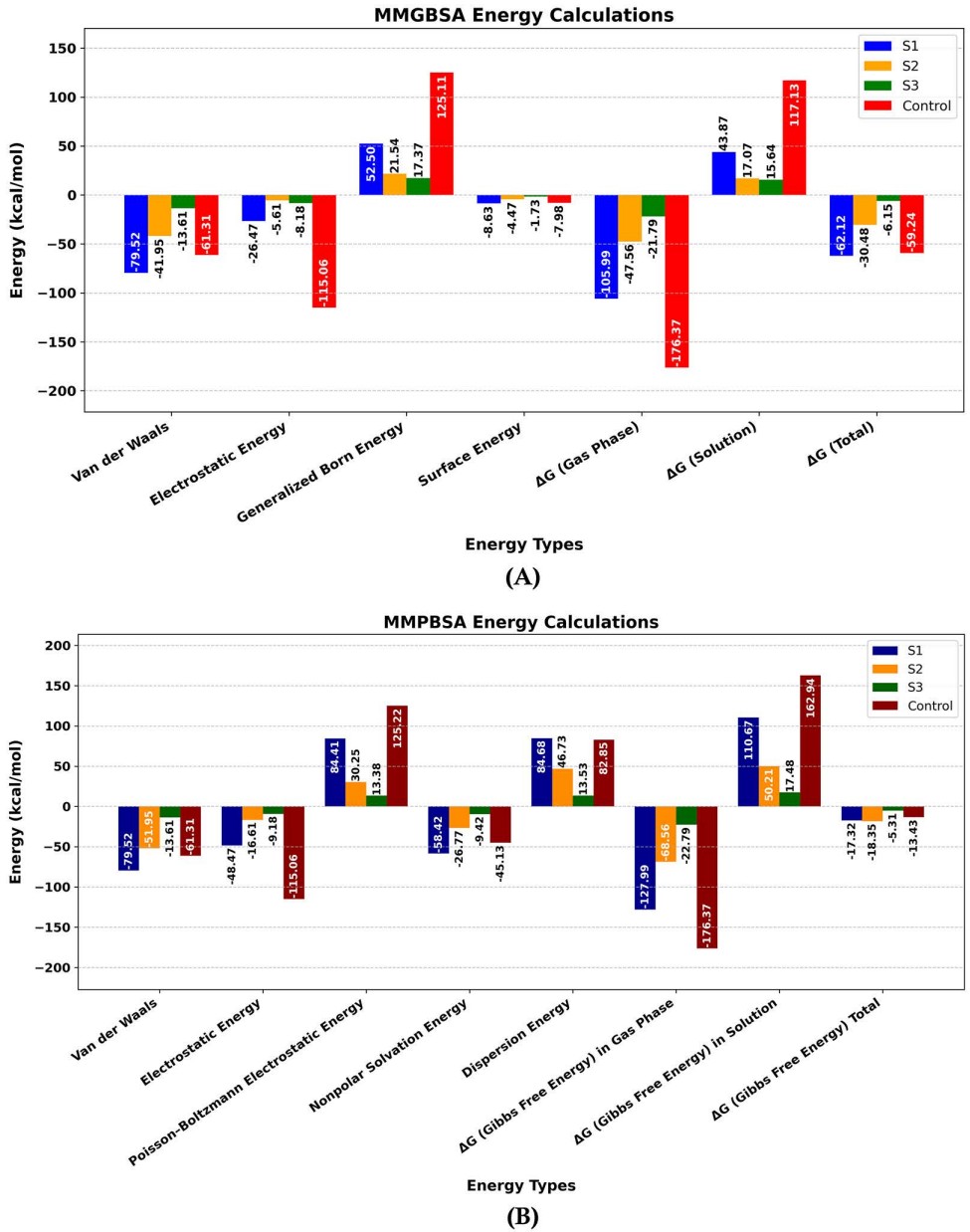

**Fig 12. Molecular mechanics (MM) of SIRT7-Compounds Complex. (A)** Molecular Mechanics Generalized Born Surface Area (MMGBSA). **(B)** Molecular Mechanics Poisson-Boltzmann Surface Area (MMPBSA) (Designed using Matplotlib 3.10.0).

an ERRAT score of 97.24% and a Verify3D score of 86.75%, establishing a robust foundation for subsequent molecular analyses. The subcellular localization studies demonstrated SIRT7's predominant nuclear localization, particularly in nucleoplasm and nuclear speckles, consistent with its role in chromatin regulation and ribosomal RNA transcription [95,96].

Gene ontology and pathway analyses suggested SIRT7 as a nuclear NAD$^+$-dependent histone deacetylase with pivotal functions in chromatin remodeling, transcriptional regulation, and its predominant localization to nuclear and nucleolar compartments reflects its established role in maintaining nucleolar structural integrity and facilitating ribosomal

RNA transcription, processes fundamental to ribosome biogenesis [96]. Functionally, SIRT7 catalyzes histone H3 and H4 deacetylation, thereby modulating transcriptional programs during cell cycle progression and mitotic exit. Significant enrichment in nicotinate and nicotinamide metabolism, coupled with NAD+ biosynthetic and tryptophan catabolic pathways, demonstrates its obligate dependence on NAD+ as a metabolic cofactor, establishing a mechanistic link between cellular bioenergetics and epigenetic transcriptional control [16,97]. Collectively, these findings position SIRT7 as a critical molecular integrator coupling metabolic status to transcriptional output, consistent with its documented roles in nucleolar homeostasis and cancer-associated metabolic reprogramming [98].

The protein-protein interaction network analysis revealed SIRT7's central role in connecting NAD+ metabolism enzymes (NADK2, NAMPT, NMNAT1, NMNAT2) with transcriptional regulators (TP53, UBTF, POLI), positioning it as a critical hub protein. Like previous studies, which implicated computational network analysis to identify key hub proteins, we have also employed computational network analysis [84,99,100]. However, we have further developed an advanced deep neural network ($R^2$: 0.9839, MSE: 0.0008, RMSE:0.0276) model that confirmed SIRT7's centrality within its specific protein cluster, with a predicted hub score of 0.674063 and the highest average combined score of 0.9358. The data showed (Table 2) a clear trend, where proteins with higher centrality values tended to have higher predicted scores, suggesting that the neural network effectively captured the complex interactions within the protein network, highlighting the utility of neural network-guided central hubs analysis for predicting protein centrality scores and understanding the topological importance of proteins within biological networks. Moreover, SIRT7's enrichment in NAD-dependent processes and histone deacetylation pathways (FDR = 5.07e-33), aligns with the previous studies [5,101], suggesting SIRT7 functions as a specialized coordinator within a defined protein network rather than a global cellular hub. In addition, cancer hallmark analysis identified genome instability as the primary pathway significantly associated with SIRT7 (adj. p-value < 0.05), while other canonical cancer hallmarks showed minimal or no association.

Furthermore, expression profiling revealed differential SIRT7 expression across 17 cancer types, with elevated expression in most tumor types except for colon adenocarcinoma and kidney chromophobe, where underexpression was observed. However, previous studies also identified that overexpression of SIRT7 is associated with both colorectal cancer and renal cell carcinoma [14,87]. The survival analysis demonstrated significant prognostic value, with high SIRT7 expression associated with poor overall survival in kidney renal clear cell carcinoma (HR = 1.6, p = 0.0016), kidney renal papillary cell carcinoma (HR = 1.9, p = 0.037), lower grade glioma (HR = 1.7, p = 0.0028), liver hepatocellular carcinoma (HR = 1.6, p = 0.0057), prostate adenocarcinoma (HR = 8.8, p = 0.04), and uterine corpus endometrial carcinoma (HR = 2.4, p = 0.023). Remarkably, sarcoma showed an inverse relationship where high SIRT7 expression was associated with better survival (HR = 0.63, p = 0.026), indicating a context-dependent, particularly tissue-specific function of SIRT7.

The analysis of SIRT7 genetic alterations reveals significant cancer-specific mutations and their downstream effects on patient outcomes and molecular pathways. The study identified multiple cancer-specific SNPs, including K72N in hepatobiliary cancer, A114V in esophagogastric cancer, Q167E and D234Y in breast cancer, H226Y, R289W, T341I, and G375S in melanoma, G246R in head and neck cancer, G268V and P368A in non-small cell lung cancer, and K395N in colorectal cancer. The survival analysis demonstrated that patients with altered SIRT7 showed decreased overall survival probability (HR = 0.734, 95% CI: 0.376–1.433). The impact of these alterations extends to significant expression changes in the SIRT7 protein network, with notable alterations including substantial downregulation of ENPP3 (Log2FC = −4.23) in COAD, NMNAT2 in SARC (Log2FC = −4.34) and STAD (Log2FC = −2.09), BST1 in BLCA (Log2FC = −1.10), ENPP1 in BLCA (Log2FC = −2.39), NMNAT2 in BLCA (Log2FC = −2.53), NNMT in CESC (Log2FC = −1.04), and upregulation of CD38 in BRCA (Log2FC = 1.20). These findings have profound implications for cancer therapeutics, suggesting that SIRT7 mutations not only affect patient survival but also significantly impact various other molecular components regulating pathways, restrengthen its context-dependent role, and potentially offer a new target for personalized treatment strategies based on specific genetic alterations and their downstream effects.

The complex interplay between SIRT7 expression and immune cell infiltration demonstrates significant cancer-specific impacts on patient survival, supported by compelling statistical evidence across multiple cancer types. In GBM, high SIRT7 expression, accompanied by elevated CD8+T cells, was associated with diminished survival (HR = 1.91, p = 0.0143), whereas KIRP exhibited an even stronger negative correlation (HR = 3.36, p = 0.0228). Notably, UVM presented a contrasting pattern where low SIRT7 expression with high CD8+T cells correlated with reduced survival (HR = 4.74, p = 0.0697). LGG displayed unique characteristics, with high CD4+T cell infiltration negatively impacting survival in both low SIRT7 expression (HR = 2.87, p = 0.000216) and high SIRT7 expression (HR = 1.9, p = 0.0132) scenarios, while B cell infiltration showed varying effects with low SIRT7 expression (HR = 1.96, p = 0.0179) and high SIRT7 expression (HR = 1.57, p = 0.059). These statistically significant findings of the modulation of SIRT7 by immune cell-specific context emphasize the need for cancer-specific therapeutic approaches that consider both SIRT7 expression levels and immune infiltrate profiles, potentially revolutionizing personalized immunotherapy strategies through more targeted interventions based on these molecular and cellular interactions.

The elucidation of SIRT7 regulatory networks reveals a complex interplay of modulatory mechanisms, underscoring the protein's multifaceted control. The identification of various drugs and compounds, including Acetaminophen, Curcumin, and Quercetin, as potential SIRT7 modulators highlights the therapeutic implications of targeting SIRT7. Furthermore, the discovery of five microRNAs (hsa-miR-17-5p, hsa-miR-34a-5p, hsa-miR-125a-5p, hsa-miR-125b-5p, and hsa-miR-335-5p) and transcription factors (FOXC1, SRF, and TFAP2A) as regulators of SIRT7 expression suggests a layered regulatory architecture, involving both transcriptional and post-transcriptional control. These findings have significant implications for the development of novel therapeutic interventions, as they provide a framework for understanding the complex regulatory mechanisms governing SIRT7 activity and identifying potential targets for modulation. The integrated network presented in S6 Fig provides a valuable resource for future studies, enabling researchers to explore the therapeutic potential of SIRT7 modulation in cancers.

Furthermore, the structural and computational analyses of SIRT7 inhibition have revealed several promising therapeutic candidates through a comprehensive evaluation of binding mechanisms and stability. The characterization of the SIRT7 active site, comprising an inhibitory region (residues 116−128) [90] and crucial catalytic residues (169, 187, 237−243, 272, 273, 277), established the foundation for structure-based drug design. Among the pharmacophore features-based screened compounds, ZINC000150487575 exhibited exceptional binding affinity (−12.5225 kcal/mol) through specific interactions with ARG120, GLY121, and ASP118 at the inhibitory site, surpassing both the control inhibitor (−9.71424 kcal/mol) and other candidates ZINC000150641215 (−10.6249 kcal/mol) and MCULE-8582664468 (−10.5185 kcal/mol). Toxicity profiling demonstrated favorable safety profiles for ZINC000150487575, ZINC000150641215, and the control inhibitor, while MCULE-8582664468 contained an aromatic amine as a potential toxicophore. Hydrogen bonding interactions common to most compounds predominantly involve residues ASP118, ARG120, and ASN123 within the SIRT7 inhibitory site. The molecular dynamics simulations revealed distinct stability patterns, with ZINC000150487575 showing superior conformational stability (RMSD: 1.5882Å) compared to ZINC000150641215 (2.9337Å), MCULE-8582664468 (4.1824Å), and the control (2.8052Å). Each compound established unique hydrogen bonding networks: ZINC000150487575 formed extensive interactions with residues 182−198, ZINC000150641215 with residues 197−199, MCULE-8582664468 with residue 193, and the control with residues 171, 178, and 180. All compounds exhibited notable hydrogen bonding interactions clustered around residues 170−200, highlighting the importance of this region for binding stability. The binding stability was further confirmed through MM-GBSA analysis, where ZINC000150487575 demonstrated the most favorable binding free energy (ΔG Total: −62.22 kcal/mol), followed by the control (−59.24 kcal/mol), ZINC000150641215 (−30.48 kcal/mol), and MCULE-8582664468 (−6.15 kcal/mol). MM-PBSA calculations corroborated these findings, with ZINC000150641215 and ZINC000150487575 showing the most favorable energetics (−18.95 and −17.32 kcal/mol, respectively) compared to the control (−13.43 kcal/mol) and MCULE-8582664468 (−5.31 kcal/mol). These comprehensive analyses

identify ZINC000150487575 and ZINC000150641215 as particularly promising lead compounds for SIRT7 inhibition, potentially offering new therapeutic strategies for various cancers where SIRT7 exhibits promotional effects, and thus establishing a framework for future experimental validation and therapeutic development efforts.

Collectively, these findings suggest that SIRT7 serves as a context-specific biomarker and personalized therapeutic target, mediating tumor progression through its integrated effects on gene regulation, mutational landscape, and immune modulation. Our findings align with previous studies demonstrating SIRT7's oncogenic role in multiple cancer types, including melanoma, non-small cell lung cancer, gastric cancer, and hepatocellular carcinoma [94]. However, our pan-cancer approach reveals several novel insights that extend beyond previous investigations. The identification of sarcoma as an exception, where high SIRT7 expression correlates with better survival, contrasts with its oncogenic role in other cancer types and suggests previously unrecognized tumor-suppressive functions in specific contexts (Fig 5H). A previous study has stated that high SIRT7 expression correlates with poor prognosis, particularly in osteosarcoma patients; therefore, contrasting our findings needs further investigation [102].

Moreover, our comprehensive analysis of protein-protein interactions provides a more detailed understanding of SIRT7's molecular environment than previous studies. The deep neural network-guided analysis represents a methodological advancement over traditional centrality measures, providing more nuanced insights into SIRT7's network position by revealing its central role in connecting NAD+ metabolism with transcriptional regulation. Additionally, the identification of cancer-specific mutations and their potential functional consequences extends previous genomic studies by systematically cataloging alterations across multiple cancer types (Fig 6A and C). In addition, the immune infiltration analysis reveals previously unexplored relationships between SIRT7 expression and immune cell populations (Fig 7), providing new insights into its role in tumor-immune interactions that were not comprehensively addressed in earlier studies.

Some limitations must be acknowledged in our comprehensive analysis. Initially, the 3D structural model, despite achieving high quality scores, remains computationally predicted rather than experimentally determined due to the current lack of reliable and valid 3D structure availability, which may impact the accuracy of subsequent molecular docking and drug discovery efforts. However, the inhibitory site information within the active site is exactly aligned with the previous studies (Fig 8A) [90,92,93]. A previous study that discovered SIRT7 inhibitors used the catalytic residues 169–277 as their binding site [92]. Another study shows the key binding site amino acids: PRO117, ASP118, ARG120, ASN168, ASP170, HIS187, and LEU274, which are aligned with our study (Table 4) [93]. The pharmacophore-based virtual screening approach, though systematic, may miss potential inhibitors due to the structure-based pharmacophore approach, which is unlike the case of the ligand-based pharmacophore approach [99,103]. However, due to the unavailability of a crystalized 3D structure, a structure-based pharmacophore hypothesis generation is crucial, and superimposition with the identified ligand to validate the pharmacophore hypothesis and features makes it strongly reliable. Additionally, the toxicity predictions are based on computational models and require extensive experimental validation to confirm their safety profiles. Furthermore, the correlation of high SIRT7 expression with better survival outcome in sarcoma needs further experimental studies to validate, and the bioactivity of identified potential inhibitors needs cell line experimental validation against diverse cancer types (Table 3) to confirm their inhibitory potential.

Despite these limitations, this study provides the most comprehensive pan-cancer analysis of SIRT7 to date, offering valuable insights that significantly advance our understanding of its role in cancer biology. The identification of SIRT7 as a prognostic biomarker across multiple cancer types has immediate clinical relevance, potentially informing treatment decisions and patient stratification strategies. Furthermore, the discovery of the central role of SIRT7 in connecting NAD+ metabolism with transcriptional regulation provides a mechanistic framework for understanding how metabolic reprogramming contributes to cancer progression. Additionally, revealing genome instability as the sole cancer hallmark offers new perspectives on SIRT7's primary oncogenic mechanism.

Apart from these, the systematic identification of regulatory networks controlling SIRT7 expression provides multiple potential therapeutic intervention points, from microRNA-based therapies to small-molecule modulators. The discovery of cancer-specific mutations offers opportunities for personalized medicine approaches, particularly in melanoma and other heavily mutated cancer types. The molecular docking and subsequent molecular dynamics validation also provide a rational foundation for novel inhibitor discovery efforts, with three compounds showing superior binding characteristics compared to the existing experimentally validated SIRT7 inhibitor. Interestingly, the binding and interaction patterns of the three newly identified inhibitors are remarkably similar to those of the control experimentally validated inhibitor, as determined by both molecular docking (Table 4 and Fig 10) and molecular dynamics simulation (Figs 11C, D, E, and F) studies. Moreover, the tissue-specific expression patterns and prognostic significance establish SIRT7 as a viable therapeutic target with potentially manageable off-target effects.

The development of SIRT7-specific biomarkers and therapeutic strategies should consider the identified regulatory networks (S6 Fig), particularly the role of microRNAs and transcription factors in controlling its expression. Future studies should also investigate the potential for combination therapies targeting both SIRT7 and its interacting partners in the NAD+ metabolism and transcriptional regulation networks, prioritizing deeper insights into allosteric regulation mechanisms of SIRT7 and their role in therapeutic resistance and disease recurrence.

## 5. Conclusion

This comprehensive pan-cancer analysis represents SIRT7 as a pivotal, context-dependent oncoprotein and prognostic biomarker, influencing tumor progression through its integrated roles in genome stability, transcriptional regulation, NAD+ metabolism, and immune modulation. Our study advances cancer biology by delivering systematic characterization of SIRT7's expression profiles, prognostic impacts (including a novel tumor-suppressive association in sarcoma that contrasts its oncogenic effects in other cancers), identifying a central hub in molecular networks using a deep neural network, cancer-specific mutation landscapes affecting survival, immune context dependencies, and multifaceted regulatory mechanisms involving targeted microRNAs and transcription factors. Further analysis of therapeutic potentials using molecular docking, molecular dynamics simulation, and post-dynamics binding free energy calculations introduces ZINC000150487575 and ZINC000150641215 as promising, computationally validated SIRT7 inhibitors. These contributions have immediate implications for clinical oncology, enabling SIRT7 as a biomarker for patient stratification. Moreover, it provides a framework for SIRT7 as a therapeutic target for personalized interventions, such as the development of novel inhibitors and the modulation of regulatory networks. While the reliance on a computationally predicted SIRT7 structure represents a limitation, its high-quality validation scores and alignment with prior structural experimental evidence make it a significant step forward in cancer biology and therapeutic interventions. The tissue-specific expression patterns and immune infiltration relationships position this research within the broader context of precision oncology and personalized medicine, where understanding the complex interplay between metabolic reprogramming, epigenetic regulation, and tumor-immune interactions becomes increasingly critical. Future investigations building upon this foundation should prioritize experimental validation of the identified inhibitors, exploration of SIRT7's role in therapeutic resistance, and development of combination strategies targeting SIRT7 interactors (e.g., NAD+ pathway elements) that leverage its central position in cellular networks to maximize therapeutic efficacy across diverse cancer contexts.

## Supporting information

**S1 Fig. Statistically significant (Green Color) association of SIRT7 with key cancer hallmarks (Source: Cancer Hallmarks Database).**
(DOCX)

**S2 Fig. Identification of protein clusters.** (A) K-Means clustering, (B) MCL Clustering, and (C) DBSCAN clustering.
(DOCX)

**S3 Fig. Automated deep learning model for 12-centrality metrics calculations.** (A) Deep learning neural network model. (B) Internal validation $R^2$ over 1000 epochs, (C) Loss of training and validation sets over 1000 epochs (generated using matplotlib 3.10.0 of Python 3.12.11).
(DOCX)

**S4 Fig. Influence of Centrality Metrics.** (A) The most important features for the deep learning model. (B) Statistical analysis of 12 centrality metrics for proteins.
(DOCX)

**S5 Fig. GO and Pathway enrichment analysis.** (A) Dot plot for GO enrichment, and (B) Bar plot for pathway enrichment analysis (generated using matplotlib 3.10.0 of Python 3.12.11).
(DOCX)

**S6 Fig. Gene regulatory network illustrating interactions among SIRT7, transcription factors, microRNAs, and modulatory compounds.**
(DOCX)

**S1 Table. Files for model development.**
(XLSX)

**S2 Table. Detailed pharmacophore features with spatial coordinates and activation status.**
(DOCX)

**S3 Table. Screening of Compounds from the publicly available databases based on pharmacophore features.**
(DOCX)

**S4 Table. Molecular docking of pharmacophore-guided screened compounds with toxicity profiling.**
(XLSX)

## Acknowledgments

The authors thank the AID Research Academy, Bangladesh, for providing technical guidance and support.

## Author contributions

**Conceptualization:** K.M. Tanjida Islam, Shahin Mahmud.

**Formal analysis:** K.M. Tanjida Islam, Shahin Mahmud.

**Investigation:** Shahin Mahmud.

**Methodology:** K.M. Tanjida Islam, Shahin Mahmud.

**Supervision:** Shahin Mahmud.

**Validation:** K.M. Tanjida Islam, Shahin Mahmud.

**Writing – original draft:** K.M. Tanjida Islam.

**Writing – review & editing:** Shahin Mahmud.

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
