## [Decision Letter · Decision Letter 0]

24 Sep 2025

Dear Dr. Mahmud,

Thank you for submitting your manuscript to PLOS ONE. After careful consideration, we feel that it has merit but does not fully meet PLOS ONE’s publication criteria as it currently stands. Therefore, we invite you to submit a revised version of the manuscript that addresses the points raised during the review process.

We look forward to receiving your revised manuscript.

Kind regards,

Firoz Ahmed

Academic Editor

PLOS ONE

Journal Requirements:

3. Please include captions for your Supporting Information files at the end of your manuscript, and update any in-text citations to match accordingly. Please see our Supporting Information guidelines for more information: http://journals.plos.org/plosone/s/supporting-information .

Additional Editor Comments:

Dear Dr. Mahmud,

Thank you for submitting your manuscript entitled “SIRT7 as a Context-Dependent Biomarker and Therapeutic Target: Insights from a Pan-Cancer Study.” The reviewers have now completed their evaluation, and I am attaching their comments. Please review these carefully and address each point in detail.

In addition to the reviewers’ feedback, I have the following editorial suggestions. These need to be implemented to strengthen and improve the quality of the manuscript:

Although the manuscript uses multiple techniques, there is insufficient rationale and justification for their selection, and in many places the analyses are not logically connected. The manuscript is large and needs to be more focused—clearly defining the core problem and the steps to address it.

Furthermore, the crystal structure of SIRT7 (or of at least parts of it) is already available. Given these existing structural data, the authors must explain why they chose to use AlphaFold to generate the SIRT7 structure in their work.

Please submit the revised manuscript along with a detailed response letter outlining how you have addressed each reviewer’s concern and the editorial comments above.

Thank you for your efforts. I look forward to receiving your revised manuscript.

Sincerely,

Academic Editor on PLOS ONE

Firoz Ahmed, PhD

Reviewer 1

Major Comments

1. Overuse of Computational Methods Without Clear Rationale

The authors employ numerous techniques—machine learning, molecular docking, pharmacophore modeling, survival analysis, network analysis, and more—without a coherent rationale or integration. This excessive use of methods appears unfocused and gives the impression of trying to include as much data as possible rather than answering a specific biological question. The manuscript would benefit from narrowing its scope and focusing on a few well-justified approaches.

2. Lack of Solid Biological Findings

Despite the extensive data analysis, the manuscript does not present any strong or novel biological insights. The results are largely descriptive, and there is minimal effort to connect computational outputs to known or hypothesized biological mechanisms involving SIRT7. For example, enriched GO terms such as rRNA processing are mentioned but not explained in the context of SIRT7’s established role in nucleolar function and ribosome biogenesis.

3. No Experimental Validation

The manuscript proposes SIRT7 as a therapeutic target but does not include any experimental data to support this claim. Validation in biological systems (e.g., cell lines, patient samples) is essential to substantiate computational findings and enhance translational relevance.

4. Unsubstantiated Context-Dependent Role

Although the title emphasizes SIRT7’s context-dependent behavior, the manuscript does not provide sufficient comparative analysis or mechanistic explanation to support this claim. The context-dependence remains an untested assertion.

Minor Comments

• Poor Figure Resolution Across the Manuscript

All figures suffer from low resolution and poor formatting, making them difficult to read and interpret. Text is often blurry or too small, and visual elements lack clarity. The authors should regenerate all figures at high resolution, with appropriate font sizes, labeling, and formatting suitable for publication.

• Improper Attribution in Figure 3b

Figure 3b appears to include an image from the Human Protein Atlas, but this is not acknowledged in the figure legend or main text. The authors must properly cite the source and ensure compliance with licensing terms.

• Language and Style

The manuscript would benefit from thorough language editing to improve grammar, flow, and scientific tone.

• Discussion of Limitations

The authors should explicitly acknowledge the limitations of relying solely on in silico analyses and public datasets, including potential biases and lack of experimental control.

Reviewer 2

I have a few minor concerns as outlined below:

1. The crystal structure of SIRT7 is already available (e.g., PDB ID: 9GMK), albeit in complex with DNA and histones. Why was there a need to model the full protein instead of utilizing or modifying the available structure?

2. While identifying the active site residues, which specific function of the protein are you considering? How would inhibition of that function contribute to cancer management or therapy?

3. In Figure 15, you mention inhibitory and catalytic residues; however, only a single reference (Reference 80, for inhibitory residues) is cited, and without sufficient detail. Please provide more information from this reference and, if available, cite additional literature supporting the identified active site residues.

4. All compounds selected for molecular docking have molecular weights greater than 500, thereby violating the first criterion of Lipinski’s Rule of Five. How do you justify the druggability of these compounds?

5. The toxicity analysis appears incomplete. Please expand it to include other key parameters. In addition, provide a complete tabulation of Lipinski’s Rule of Five compliance for the selected compounds.

6. Are there any previous reports of small molecules or chemical compounds being identified as inhibitors of SIRT7? If so, please cite and discuss them.

7. Kindly overlay each of the selected docking compounds onto the pharmacophore model individually to illustrate their fit.

8. Please remove retracted articles from the reference list (e.g., Reference No. 88) and strengthen the manuscript by citing more recent and relevant publications.

9. You mention that high SIRT7 expression correlates with poor survival, particularly in sarcoma. What do previous reports state on this correlation, and how consistent are your findings with existing literature?

Editor’s Comments on the Manuscript

1. Logical Flow & Conciseness

As one reviewer pointed out, the Results section appears disjointed and lacks logical continuity. The manuscript should be reorganized so that each analysis builds on the previous one in a clear scientific progression. Additionally, the manuscript is quite long; please trim unnecessary parts and present results more concisely.

2. Background / Function of SIRT7 under Normal Conditions

The manuscript currently lacks sufficient description of SIRT7’s role in normal (non-cancer) biology. Please include a section that outlines the normal function(s) and mode(s) of action of SIRT7, with appropriate references.

3. Gene vs Protein Nomenclature

Ensure that gene names are in italics (e.g. SIRT7) while protein names are in non-italics. This must be consistent throughout the manuscript (title, abstract, introduction, results, discussion).

4. References & Citation Style

Use the PLOS ONE reference style

5. Unreferenced Statements in Introduction

For example, the sentence:

“SIRT7 is considered an oncogene, as evidenced by Hepatocellular carcinoma (HCC), where the overexpression of SIRT7 in human HCC samples increased with tumor grade.”

This statement needs a proper citation. Please add references that support this assertion.

6. Choice and Logic of Interaction Tools

You used multiple tools/databases to identify interacting partners (GeneMANIA, STRING, BioGRID). Please explain clearly:

o Why you selected more than one tool.

o What the principle/algorithm behind each tool is.

o What type of interactions each tool reveals (e.g. physical binding, genetic interaction, co-expression).

7. Methods / Datasets

The methodology section needs clearer description of all datasets used, including origin, versions, preprocessing steps. For section 2.2.4 (“Automated Central Hub Identification”), describe in detail how central hubs are defined, the metrics used, and how thresholds were determined.

8. Choice of Tools for Expression vs Survival Analysis

In section 2.3.1, you use TIMER2 for gene expression analysis and GEPIA2 for survival analysis. Explain why different tools were used for these related analyses. Could using GEPIA2 for both provide consistency? Clarify the rationale.

9. Results vs Discussion Balance

In subsection 3.1.1, the manuscript presents:

“Fig 2. represents the structural quality assessment of the SIRT7 3D structure.”

But then there is no further discussion of what the findings mean. Please interpret figure results: what metrics in Fig 2 indicate good vs poor structure quality? What impact does this have for your downstream analysis?

10. Figure Legends & Tool Attribution

All figure legends must clearly state which tool / software / database was used to generate the figure. Specifically for Figure 6, 7, 8: you must indicate in each legend whether the figure was produced using STRING, GeneMANIA, etc.

11. Deep Neural Network Section Needs More Detail

The section 3.2.4 “Deep Neural Network Guided Central Hubs” is missing critical methodological details. Please include:

o What datasets were used.

o What features (independent variables) and labels/outcomes (dependent variables) were used.

o How the model was trained, validated, tested.

o Performance metrics.

o If possible, share the code (e.g. via GitHub) so that your analyses can be replicated.

Reviewers' comments:

Reviewer's Responses to Questions

**Comments to the Author**

1. Is the manuscript technically sound, and do the data support the conclusions?

Reviewer #1: Yes

Reviewer #2: Yes

2. Has the statistical analysis been performed appropriately and rigorously?

Reviewer #1: I Don't Know

Reviewer #2: Yes

3. Have the authors made all data underlying the findings in their manuscript fully available?

Reviewer #1: Yes

Reviewer #2: Yes

4. Is the manuscript presented in an intelligible fashion and written in standard English?

Reviewer #1: Yes

Reviewer #2: Yes

Reviewer #1: The manuscript attempts to explore the role of SIRT7 across multiple cancer types using a wide array of computational approaches. While the topic is relevant and the authors demonstrate enthusiasm in covering diverse analytical angles, the study lacks a clear biological focus, methodological novelty, and scientific rigor. The manuscript reads more like a collection of disconnected analyses rather than a hypothesis-driven investigation.

Major Comments

1. Overuse of Computational Methods Without Clear Rationale

The authors employ numerous techniques—machine learning, molecular docking, pharmacophore modeling, survival analysis, network analysis, and more—without a coherent rationale or integration. This excessive use of methods appears unfocused and gives the impression of trying to include as much data as possible rather than answering a specific biological question. The manuscript would benefit from narrowing its scope and focusing on a few well-justified approaches.

2. Lack of Solid Biological Findings

Despite the extensive data analysis, the manuscript does not present any strong or novel biological insights. The results are largely descriptive, and there is minimal effort to connect computational outputs to known or hypothesized biological mechanisms involving SIRT7. For example, enriched GO terms such as rRNA processing are mentioned but not explained in the context of SIRT7’s established role in nucleolar function and ribosome biogenesis.

3. No Experimental Validation

The manuscript proposes SIRT7 as a therapeutic target but does not include any experimental data to support this claim. Validation in biological systems (e.g., cell lines, patient samples) is essential to substantiate computational findings and enhance translational relevance.

4. Unsubstantiated Context-Dependent Role

Although the title emphasizes SIRT7’s context-dependent behavior, the manuscript does not provide sufficient comparative analysis or mechanistic explanation to support this claim. The context-dependence remains an untested assertion.

Minor Comments

• Poor Figure Resolution Across the Manuscript

All figures suffer from low resolution and poor formatting, making them difficult to read and interpret. Text is often blurry or too small, and visual elements lack clarity. The authors should regenerate all figures at high resolution, with appropriate font sizes, labeling, and formatting suitable for publication.

• Improper Attribution in Figure 3b

Figure 3b appears to include an image from the Human Protein Atlas, but this is not acknowledged in the figure legend or main text. The authors must properly cite the source and ensure compliance with licensing terms.

• Language and Style

The manuscript would benefit from thorough language editing to improve grammar, flow, and scientific tone.

• Discussion of Limitations

The authors should explicitly acknowledge the limitations of relying solely on in silico analyses and public datasets, including potential biases and lack of experimental control.

Reviewer #2: I have a few minor concerns as outlined below:

1. The crystal structure of SIRT7 is already available (e.g., PDB ID: 9GMK), albeit in complex with DNA and histones. Why was there a need to model the full protein instead of utilizing or modifying the available structure?

2. While identifying the active site residues, which specific function of the protein are you considering? How would inhibition of that function contribute to cancer management or therapy?

3. In Figure 15, you mention inhibitory and catalytic residues; however, only a single reference (Reference 80, for inhibitory residues) is cited, and without sufficient detail. Please provide more information from this reference and, if available, cite additional literature supporting the identified active site residues.

4. All compounds selected for molecular docking have molecular weights greater than 500, thereby violating the first criterion of Lipinski’s Rule of Five. How do you justify the druggability of these compounds?

5. The toxicity analysis appears incomplete. Please expand it to include other key parameters. In addition, provide a complete tabulation of Lipinski’s Rule of Five compliance for the selected compounds.

6. Are there any previous reports of small molecules or chemical compounds being identified as inhibitors of SIRT7? If so, please cite and discuss them.

7. Kindly overlay each of the selected docking compounds onto the pharmacophore model individually to illustrate their fit.

8. Please remove retracted articles from the reference list (e.g., Reference No. 88) and strengthen the manuscript by citing more recent and relevant publications.

9. You mention that high SIRT7 expression correlates with poor survival, particularly in sarcoma. What do previous reports state on this correlation, and how consistent are your findings with existing literature?

**Do you want your identity to be public for this peer review?** For information about this choice, including consent withdrawal, please see our Privacy Policy

Reviewer #1: No

Reviewer #2: **Yes:** Mohd Rehan

---

## [Author Response · Author response to Decision Letter 1]

7 Nov 2025

05 November, 2025

Firoz Ahmed

Academic Editor

PLOS ONE

Dear Editor,

Special gratitude to you for allowing us to resubmit a more revised and updated version of our manuscript entitled "SIRT7 as a Context-Dependent Biomarker and Therapeutic Target: Insights from a Pan-Cancer Study" (Manuscript ID: PONE-D-25-42611) to the PLOS ONE journal as an original research article. We have organized the manuscript based on the editor’s and reviewers’ comments. In response to valuable suggestions, we have modified, revised, organized, corrected, and narrowed the manuscripts. The result section is now presented with a clear rationale for each step, and a logical flow has been maintained. Some figures have now been presented as supplementary figures to make the manuscript more concise. However, due to it being a pan-cancer study, we tried to provide logical and relevant information as much as possible, and all the limitations have been addressed in the discussion section. We also provided some logical explanations of particular queries of the honorable reviewers and editor, particularly regarding the use of the AlphaFold3 modeled structure rather than the crystal structure. Additionally, we have appropriately addressed all queries of reviewers in the response letter, line by line. We hope our response below will successfully address the reviewer’s specific points and improve the manuscript. Please let us know if you have any additional questions or comments. Finally, we would like to express our thanks and gratitude for your kind cooperation.

Sincerely yours,

Shahin Mahmud

Assistant Professor

Department of Biotechnology and Genetic Engineering

Mawlana Bhashani Science and Technology University, Santosh, Tangail-1902, Bangladesh.

Email: shahin018mbstu@gmail.com

Responses to the honorable reviewers’ s comments

Additional Editor Comments:

Dear Dr. Mahmud,

Thank you for submitting your manuscript entitled “SIRT7 as a Context-Dependent Biomarker and Therapeutic Target: Insights from a Pan-Cancer Study.” The reviewers have now completed their evaluation, and I am attaching their comments. Please review these carefully and address each point in detail.

In addition to the reviewers’ feedback, I have the following editorial suggestions. These need to be implemented to strengthen and improve the quality of the manuscript:

Comments/Questions/Instruction/Guidelines:

Although the manuscript uses multiple techniques, there is insufficient rationale and justification for their selection, and in many places the analyses are not logically connected. The manuscript is large and needs to be more focused—clearly defining the core problem and the steps to address it.

Responses of the Authors:

We have now clearly explained and justified the rationale for the selection of each technique and connected them to focus on justifying the title of our manuscript.

Specifically, we have now focused on

(1) Determining the biomarker possibility of SIRT7 by network, expression, survival, alteration, and immune association analysis.

(2) exploring therapeutic opportunities by identifying active sites, inhibitory sites, molecular docking, molecular dynamics, and post-dynamics MMPBSA studies.

To justify our study as a pan-cancer study, we have to use several techniques, which we have clarified now. Therefore, the manuscript is large. However, we have trimmed our manuscript by focusing on the core problem and keeping the steps sequential that are required to address the problem. Minor or less important techniques have been merged or excluded, and some repetitive or less important figures are presented now as supplementary figures.

We have designed this study in a way that researchers who are interested in SIRT7 may find valuable information and insights for subsequent experiments.

Comments/Questions/Instruction/Guidelines:

Furthermore, the crystal structure of SIRT7 (or of at least parts of it) is already available. Given these existing structural data, the authors must explain why they chose to use AlphaFold to generate the SIRT7 structure in their work.

Responses of the Authors:

Thank you for raising this issue. Although experimental structures are available, we did not find them suitable for our study. Before starting our study, we have thoroughly reviewed the structure of SIRT7 to identify any crystal structure of SIRT7 with co-crystallized ligand binding at the inhibitory site. However, no complete crystal structure of SIRT7 is available in the PDB database. Though some fragments are available, they are not suitable for subsequent molecular docking analysis due to the lack of a proper inhibitory and catalytic site.

Additionally, the mentioned structure, 9GMK, has been resolved using Electron Microscopy and does not provide any information about inhibitory sites. Therefore, we have queried the literature to identify the inhibitory site information and presented it in Figure 11.

We have performed molecular docking targeting both the Electron Microscopic structure and the modeled structure with the experimentally validated SIRT7 inhibitor (PubChem CID: 155513088).

In terms of docking, the AlphaFold3 modeled structure suppressed the EM structure that is available at legacy PDB format: 9GMK; however, the resolution is lower (3.5 Å), and that was reflected in docking studies, where we have observed a lower binding affinity in the case of the EM structure than the AlphaFold3 model. Most interestingly, the inhibitors bind perfectly to the modeled structure at both the literature-based identified inhibitory sites and active sites (Figure 11, Table 6, and Figure 13). This specific binding and interaction pattern was possible due to the development of our pharmacophore model, which was developed based on amino acid residues of both inhibitory sites and catalytic sites of the main pocket.

Earlier, we stated the unavailability of a crystal structure. Although some fragments of the crystal structure are available, they are not suitable to perform docking. The resolution of the EM structure is also not high (3.5Å) and exhibited much inferior performance than our modeled structure in terms of binding affinity. Therefore, the Alphaold3 model of SIRT7 structure has been used in this study. However, we hope our study will instigate proteomics researchers to resolve the complete crystal structure of SIRT7.

We have mentioned the limitations of the modeled structure and recommend resolving the complete crystal structure of SIRT7 in the discussion section.

Thank you for guiding us to improve our manuscript.

Honorable Reviewer 1:

Major Comments

Comments/Questions/Instruction/Guidelines:

Overuse of Computational Methods Without Clear Rationale

The authors employ numerous techniques—machine learning, molecular docking, pharmacophore modeling, survival analysis, network analysis, and more—without a coherent rationale or integration. This excessive use of methods appears unfocused and gives the impression of trying to include as much data as possible rather than answering a specific biological question. The manuscript would benefit from narrowing its scope and focusing on a few well-justified approaches.

Responses of the Authors:

Thank you for addressing this issue. The reason behind choosing several techniques is that it is a pan-cancer study, which required multidimensional analysis to strengthen and support findings.

We have reviewed the entire manuscript and have now justified the rationale for integrating these techniques at each step. However, we have narrowed our scope by excluding minor, less impactful, or repetitive techniques as per your suggestions.

In the revised manuscript, we have particularly focused on justifying the biomarker and therapeutic opportunities, but still tried to provide most of the relevant information with a clear rationale to justify our pan-cancer study.

We have primarily divided our study into 3 parts:

1. Network analysis to determine how important SIRT7 is in the biological network of SIRT7 partners. We found SIRT7 is the central hub among their partners.

2. Determines the Context-dependent role of SIRT7 in cancer using gene expression, survival, mutational, and immune association studies, where we have found that the overexpression of SIRT7 is responsible for several cancers, supported by our literature review study (Table 3). However, in some cancer types, underexpression is also related to cancer progression, such as the underexpression of SIRT7 is responsible for the progression of breast cancer [1,2].

3. Identification of novel inhibitors using computational approaches, where we have identified 3 promising compounds.

Comments/Questions/Instruction/Guidelines:

Lack of Solid Biological Findings

Despite the extensive data analysis, the manuscript does not present any strong or novel biological insights.

Responses of the Authors:

Thank you for addressing this query. Our study aims to present the context-dependent role of SIRT7 in different types of cancers, as we found that the role of SIRT7 is not straightforward in cancer. In some cancer types, the overexpression of SIRT7 has been observed, and in some cases, underexpression. We have found that the overexpression of SIRT7 is better for the survival of sarcoma, and because this is a novel finding of our study therefore we have recommended the requirement of further experimental validation to confirm these findings. Moreover, only a single previous study presents that the overexpression of SIRT7 is related to a poor outcome for the survival of osteosarcoma patients. However, no specific studies have been conducted for soft tissue sarcoma, and therefore, due to this gap in the currently available literature, the requirement for more rigorous studies has been suggested in our study.

Besides, we have identified 3 promising novel inhibitors using computational approaches: ZINC000150487575, ZINC000150641215, and MCULE-8582664468.

Comments/Questions/Instruction/Guidelines:

The results are largely descriptive, and there is minimal effort to connect computational outputs to known or hypothesized biological mechanisms involving SIRT7. For example, enriched GO terms such as rRNA processing are mentioned but not explained in the context of SIRT7’s established role in nucleolar function and ribosome biogenesis.

Responses of the Authors:

We have now revised our manuscript to show how computational findings can be connected to known biological mechanisms, particularly in the case of GO and pathway enrichment analysis. We have addressed and highlighted this in the discussion section of our manuscript.

SIRT7 exhibits critical regulatory functions in chromatin remodeling and transcriptional modulation, and is predominantly localized within nuclear and nucleolar compartments, as indicated by the identified GO terms. This enzyme maintains nucleolar structural integrity and facilitates ribosomal RNA transcription [3]. Through its catalytic activity on histones H3 and H4, SIRT7 orchestrates transcriptional programs essential for cell cycle progression. The enzyme's requisite NAD⁺ dependence establishes a fundamental connection between cellular bioenergetics and epigenetic regulation [4,5], highlighting its significance in nucleolar homeostasis and metabolic reprogramming in oncogenic contexts [6].

Comments/Questions/Instruction/Guidelines:

No Experimental Validation

The manuscript proposes SIRT7 as a therapeutic target but does not include any experimental data to support this claim. Validation in biological systems (e.g., cell lines, patient samples) is essential to substantiate computational findings and enhance translational relevance.

Responses of the Authors:

In this case, we have designed our study based on the below:

1. We have chosen a reference potent inhibitor of SIRT7 from previously experimentally validated evidence.

2. Then, we have compared this reference inhibitor throughout our study to provide comparative insights on SIRT7 inhibitory capability.

Due to current resource and budget limitations, we are unable to perform further experimental validation. However, because our findings are promising, supported by both molecular docking and dynamics simulation studies, the dissemination of our findings will inspire other researchers to perform experimental studies who have no resources or budget limitations.

Comments/Questions/Instruction/Guidelines:

Unsubstantiated Context-Dependent Role

Although the title emphasizes SIRT7’s context-dependent behavior, the manuscript does not provide sufficient comparative analysis or mechanistic explanation to support this claim. The context-dependence remains an untested assertion.

Responses of the Authors:

Thank you for addressing this issue. In the revised manuscript, we have particularly addressed why we have stated a context-dependent role.

We have provided a detailed table summarizing the specific mechanisms by which SIRT7 is involved in cancer progression, based on our literature review analysis of recently published literature (Table 3). This analysis supports our subsequent study of exploring novel inhibitors for SIRT7.

Additionally, our computational analysis suggests that SIRT7 expression is not straightforward; rather, it is overexpressed in certain cancer types and underexpressed in certain cancer types, which provides insight into the context-dependent role of SIRT7. These computational findings are also supported by the recent literature, and some contradictory findings are also observed. Moreover, previous studies identified that the underexpression of SIRT7 is responsible for the progression of breast cancer [1,2].

This observation reinforces the requirement of this pan-cancer study to clarify the context-dependent role of SIRT7, where both similar and contradictory findings have been observed. Here, we have performed the gene expression analysis, survival analysis, mutation impact, and immune association analysis to address that the role of SIRT7 is context-dependent and its expression can vary in cancer types and can be modulated by mutational impact, CD8+ immune cell impact.

Hence, our study could guide researchers interested in SIRT7’s involvement in diverse cancer types.

However, indeed, we did not particularly focus on mechanistic insight. It is possible to conduct a standalone mechanistic study on SIRT7. In this pan-cancer study, we focus on broad aspects of cancer. We have now clarified and highlighted the context-dependent issue more precisely in the results and discussion section of the manuscript.

Comments/Questions/Instruction/Guidelines:

Please improve the quality and resolution of discussed figures.

Responses of the Authors:

Thank you for your suggestion. All figures in the manuscript have now been provided with improved quality and higher resolution (>300 dpi).

Minor Comments

Comments/Questions/Instruction/Guidelines:

Poor Figure Resolution Across the Manuscript

All figures suffer from low resolution and poor formatting, making them difficult to read and interpret. Text is often blurry or too small, and visual elements lack clarity. The authors should regenerate all figures at high resolution, with appropriate font sizes, labeling, and formatting suitable for publication.

Responses of the Authors:

We have thoroughly checked all the figures to identify figures with low quality and resolution. We have now ensured all figures in the manuscript with improved quality and higher resolution, which is suitable for publication.

Comments/Questions/Instruction/Guidelines:

Improper Attribution in Figure 3b

Figure 3b appears to include an image from the Human Protein Atlas, but this is not acknowledged in the figure legend or main text. The authors must properly cite the source and ensure compliance with licensing terms.

Responses of the Authors:

We have read the citation and license agreement of Human Protein Atlas and fulfill the criteria to use the resources from HPA by citing one primary source and all secondary sources.

Comments/Questions/Instruction/Guidelines:

Language and Style

The manuscript would benefit from thorough language editing to improve grammar, flow, and scientific tone.

Responses of the Authors:

The manuscript has be

---

## [Decision Letter · Decision Letter 1]

17 Dec 2025

Dear Dr. Dr. Mahmud,

Thank you for submitting your manuscript to PLOS ONE. After careful consideration, we feel that it has merit but does not fully meet PLOS ONE’s publication criteria as it currently stands. Therefore, we invite you to submit a revised version of the manuscript that addresses the points raised during the review process.

We look forward to receiving your revised manuscript.

Kind regards,

Firoz Ahmed

Academic Editor

PLOS One

Journal Requirements:

Additional Editor Comments:

Dear Dr. Mahmud,

Thanks for the submitting the revised manuscript and addressing the comments. Please address the comments of Reviewer #2. Please make sure the length of manuscript adhere to the journal guidelines. I found that the manuscript is still very long with 15 Figures and 7 Tables.

Sincerely,

Firoz Ahmed, PhD

Reviewers' comments:

Reviewer's Responses to Questions

**Comments to the Author**

Reviewer #1: All comments have been addressed

Reviewer #2: (No Response)

2. Is the manuscript technically sound, and do the data support the conclusions?

Reviewer #1: Yes

Reviewer #2: Yes

3. Has the statistical analysis been performed appropriately and rigorously?

Reviewer #1: Yes

Reviewer #2: I Don't Know

4. Have the authors made all data underlying the findings in their manuscript fully available?

Reviewer #1: Yes

Reviewer #2: Yes

5. Is the manuscript presented in an intelligible fashion and written in standard English?

Reviewer #1: Yes

Reviewer #2: Yes

Reviewer #1: The authors have addressed all previous comments, and the revised manuscript appears well-prepared for publication.

Reviewer #2: 1. While responding to reviewers, please highlight the added text in the revised manuscript and clearly mention the highlighted section and paragraph where the revisions have been made. It is also advisable to include page numbers and line numbers in the revised manuscript and refer to them accordingly in your responses.

2. In response to my earlier comment, “The crystal structure of SIRT7 is already available ......”, you stated: “We have performed molecular docking targeting both the Electron Microscopic structure and the modeled structure with the experimentally validated SIRT7 inhibitor (PubChem CID: 155513088).” However, I do not find any docking results for the EM structure in the revised manuscript. Furthermore, selecting a cyclic tripeptide as the reference inhibitor for docking against small-molecule compounds is not rational, given that several small-molecule chemical inhibitors of SIRT7 are already reported. Please consult the following studies and perform a comprehensive literature review:

https://www.sciencedirect.com/science/article/abs/pii/S1093326325002360

https://www.frontiersin.org/journals/cell-and-developmental-biology/articles/10.3389/fcell.2021.813233/full

https://www.sciencedirect.com/science/article/abs/pii/S0006291X1832552X

3.Your response to the comment “While identifying the active site residues .......” is not satisfactory. Relevant data and previously designed inhibitors are available. Please refer to the studies mentioned above and revise your response accordingly.

4. Please verify, based on the above studies, whether you are targeting the same binding site or a different one. In either case, you must discuss your results in the context of these previously published findings.

**Do you want your identity to be public for this peer review?** For information about this choice, including consent withdrawal, please see our Privacy Policy

Reviewer #1: No

Reviewer #2: No

---

## [Author Response · Author response to Decision Letter 2]

28 Dec 2025

27 December, 2025

Firoz Ahmed

Academic Editor

PLOS One

Dear Editor,

Special gratitude to you for allowing us to resubmit a more revised and updated version of our manuscript entitled "SIRT7 as a Context-Dependent Biomarker and Therapeutic Target: Insights from a Pan-Cancer Study" (Manuscript ID: PONE-D-25-42611R1) to the PLOS ONE journal as an original research article. We have organized the manuscript based on the editor’s and reviewers’ comments. In response to valuable suggestions, we have modified, revised, organized, corrected, and narrowed the manuscripts. We have appropriately addressed all queries of Reviewer 2 in the response letter, line by line. We hope our response below will successfully address the reviewer’s specific points and improve the manuscript. Please let us know if you have any additional questions or comments. Finally, we would like to express our thanks and gratitude for your kind cooperation.

Sincerely yours,

Shahin Mahmud

Assistant Professor

Department of Biotechnology and Genetic Engineering

Mawlana Bhashani Science and Technology University, Santosh, Tangail-1902, Bangladesh.

Email: shahin018mbstu@gmail.com

Responses to the honorable reviewers’ s comments

Honorable Reviewer 2:

Comments/Questions/Instructions/Guidelines:

While responding to reviewers, please highlight the added text in the revised manuscript and clearly mention the highlighted section and paragraph where the revisions have been made. It is also advisable to include page numbers and line numbers in the revised manuscript and refer to them accordingly in your responses.

Responses of the Authors:

Thank you for your valuable advice. We have added page numbers and line numbers in our manuscript. Also, in future submissions, we will adhere to your advice.

Comments/Questions/Instructions/Guidelines:

In response to my earlier comment, “The crystal structure of SIRT7 is already available ......”, you stated: “We have performed molecular docking targeting both the Electron Microscopic structure and the modeled structure with the experimentally validated SIRT7 inhibitor (PubChem CID: 155513088).” However, I do not find any docking results for the EM structure in the revised manuscript. Furthermore, selecting a cyclic tripeptide as the reference inhibitor for docking against small-molecule compounds is not rational, given that several small-molecule chemical inhibitors of SIRT7 are already reported. Please consult the following studies and perform a comprehensive literature review:

https://www.sciencedirect.com/science/article/abs/pii/S1093326325002360

https://www.frontiersin.org/journals/cell-and-developmental-biology/articles/10.3389/fcell.2021.813233/full

https://www.sciencedirect.com/science/article/abs/pii/S0006291X1832552X

Responses of the Authors:

Thank you for this important concern. We did not included the docking of EM structure earlier due to the superior performance of our modeled structure. However, we have included the docking results of all pharmacophore-based screened compounds for both the modeled structure and EM structure in the Supplementary Table S4 Column 4.

Again, thank you for your suggestion of this relevant literature. In our previous Supplementary Table S2 (Currently Supplementary Table S4), we already provided 2 experimentally validated SIRT7 inhibitors. One is based on the third literature you have suggested (https://www.sciencedirect.com/science/article/abs/pii/S0006291X1832552X). The control inhibitor we have used was named SIRT7 inhibitor 97491 (PubChem CID: 146018038). Due to a lower performance than the cyclic tripeptide ligand, we did not highlight it earlier. In the revised Supplementary Table S2, we have highlighted both the control inhibitors in Pink Color at the end of the Table. We have prioritized the cyclic tripeptide SIRT7 Inhibitor due to its greater binding potential with SIRT7 than other SIRT7 inhibitors (PubChem CID: 146018038). Moreover, the inhibitory potential of the cyclic tripeptide SIRT7 inhibitor is highly potent in the previous cell-line assay study. Therefore, earlier we had chosen both but prioritized the cyclic tripeptide SIRT7 inhibitor later.

Comments/Questions/Instructions/Guidelines:

Your response to the comment “While identifying the active site residues .......” is not satisfactory. Relevant data and previously designed inhibitors are available. Please refer to the studies mentioned above and revise your response accordingly.

Responses of the Authors:

Thank you for the guidance and clarification. Previously, we thought you had suggested adding only experimental evidence of inhibitory sites. Now, we have added relevant references for active site residues in the manuscript: Page: 28, Line: 11, and in the discussion section: Page: 45, Line: 2, where we have compared the active sites with previous studies.

Comments/Questions/Instructions/Guidelines:

Please verify, based on the above studies, whether you are targeting the same binding site or a different one. In either case, you must discuss your results in the context of these previously published findings.

Responses of the Authors:

We have targeted the same inhibitory sites that the above studies have suggested. We have now discussed and compared the active sites in our discussion section on Page: 45, Line: 2-5.

The 1st study you have suggested (https://www.sciencedirect.com/science/article/abs/pii/S1093326325002360) used the catalytic residues: 169-277, which aligned with our study (Figure 8A).

Figure 8 (A). Active site of SIRT7.

The 2nd study you have suggested (https://www.frontiersin.org/journals/cell-and-developmental-biology/articles/10.3389/fcell.2021.813233/full) utilized the almost exactly same binding site of our study: PRO117, ASP118, ARG120, ASN168, ASP170, HIS187, and LEU274.

Figure. The binding site from the previous study.

Table 4. Molecular docking analysis to capture interacting amino acids with compounds in the most suitable binding pose.

3rd study (https://www.frontiersin.org/journals/cell-and-developmental-biology/articles/10.3389/fcell.2021.813233/full) did not provide the binding site information; instead presented the IC50 of SIRT7 inhibitor 97491 (PubChem CID: 146018038) = 0.325 μM.

We have now discussed and compared the findings with previous studies in the discussion section: Page: 45, Line: 2-5.

Thank you for your invaluable guidelines and advice, which are really necessary to significantly strengthen our manuscript.

Additional Editor Comments:

Comments/Questions/Instructions/Guidelines:

Dear Dr. Mahmud,

Thanks for the submitting the revised manuscript and addressing the comments. Please address the comments of Reviewer #2. Please make sure the length of manuscript adhere to the journal guidelines. I found that the manuscript is still very long with 15 Figures and 7 Tables.

Responses of the Authors:

To reduce the length of the manuscript, we have presented some of the Figures and Tables in the Supplementary Section. Only the key Figures and Tables have now been retained in the main manuscript. All the Figures of the Manuscript have been provided in a minimum of 300 dpi in TIFF format in a zip file.

Thank you for your invaluable guidelines and advice, which are really necessary to significantly strengthen our manuscript.

---

## [Decision Letter · Decision Letter 2]

20 Jan 2026

SIRT7 as a Context-Dependent Biomarker and Therapeutic Target: Insights from a Pan-Cancer Study

PONE-D-25-42611R2

Dear Dr. Mahmud,

We’re pleased to inform you that your manuscript has been judged scientifically suitable for publication and will be formally accepted for publication once it meets all outstanding technical requirements.

Kind regards,

Firoz Ahmed

Academic Editor

PLOS One

Additional Editor Comments (optional):

Reviewers' comments:

Reviewer's Responses to Questions

**Comments to the Author**

Reviewer #2: All comments have been addressed

2. Is the manuscript technically sound, and do the data support the conclusions?

Reviewer #2: Yes

3. Has the statistical analysis been performed appropriately and rigorously?

Reviewer #2: I Don't Know

4. Have the authors made all data underlying the findings in their manuscript fully available?

Reviewer #2: Yes

5. Is the manuscript presented in an intelligible fashion and written in standard English?

Reviewer #2: Yes

Reviewer #2: (No Response)

**Do you want your identity to be public for this peer review?** For information about this choice, including consent withdrawal, please see our Privacy Policy

Reviewer #2: No

---

## [Editor Report · Acceptance letter]

PONE-D-25-42611R2

PLOS One

Dear Dr. Mahmud,

I'm pleased to inform you that your manuscript has been deemed suitable for publication in PLOS One. Congratulations! Your manuscript is now being handed over to our production team.

Kind regards,

on behalf of

Dr. Firoz Ahmed

Academic Editor

PLOS One